# ER-AAE: A QUANTUM STATE PREPARATION APPROACH BASED ON ENTROPY REDUCTION

## ABSTRACT

Amplitude encoding of classical vectors serves as a cornerstone for numerous quantum machine learning algorithms in real-world applications. Nevertheless, achieving exact amplitude encoding for general vectors needs an exponential number of gates, which negates the potential quantum advantages. To address the challenge of large gate number in the state preparation phase, we propose an approximate amplitude encoding algorithm based on entropy reduction (ER-AAE) within the classical framework. Given a target vector, the ER-AAE algorithm generates a sequence of gates, comprising single-qubit rotations and CZ gates, that approximates the amplitude encoding of the target vector. The structure of encoding circuits in ER-AAE is built inductively using a greedy search strategy that maximally reduces the linear entropy. We further prove that the state produced by ER-AAE approximates to the target state with the infidelity bounded by the linear entropy of intermediate states. Experimental results, including state preparations on random quantum circuit states, random vectors, MNIST digits, and CIFAR-10 images, validate our method. Specifically, real-world data reveals a noteworthy trend where linear entropy decays significantly faster compared to random vectors. Furthermore, the ER-AAE algorithm surpasses the best existing encoding techniques, achieving lower error with an equivalent or fewer number of CNOT or CZ gates.

## 1 INTRODUCTION

The experimental advancements of quantum devices (Arute et al., 2019; Wu et al., 2021; Madsen et al., 2022) have catalyzed the development of quantum algorithms (Montanaro, 2016) across various fields, including machine learning (Biamonte et al., 2017; Schuld & Killoran, 2019; Wiebe et al., 2012), quantum simulations (Daley et al., 2022), and numerical analysis (Harrow et al., 2009; Montanaro & Pallister, 2016; Xin et al., 2020; Liu et al., 2021; Lubasch et al., 2020). For practical tasks, a state preparation process is often required to encode classical data into quantum states. For instance, the quantum linear system solver (Harrow et al., 2009) (HHL algorithm) generates the solution to $A\boldsymbol{x} = \boldsymbol{b}$ in state form given the oracle preparing the state $|\boldsymbol{b}\rangle = \sum_i b_i |i\rangle / \|\boldsymbol{b}\|_2$, i.e. the amplitude encoding (Grover, 2000) of $\boldsymbol{b}$. Numerous quantum machine learning algorithms share this requirement, including quantum support vector machines (Rebentrost et al., 2014) and quantum $k$-means algorithms (Kerenidis et al., 2019; Zardini et al., 2024), which rely on the HHL algorithm as a core component. Similar demands arise in solving differential equations, where quantum solvers (Xin et al., 2020; Liu et al., 2021; Lubasch et al., 2020) require amplitude encoding for boundary conditions. Moreover, many quantum algorithms claim speed-up by focusing on oracle query complexities (Aaronson, 2015), making efficient oracle implementations essential for realizing practical quantum advantages. Specifically, quantum state preparation methods that scale polynomially with the number of qubits $N$ are highly desirable.

Despite its critical role in numerous quantum algorithms, state preparation for amplitude encodings remains computationally inefficient in general. Both exact and approximate amplitude encoding typically require $\tilde{\mathcal{O}}(2^N)$ single- and two-qubit gates (Grover, 2000; Kaye & Mosca, 2001; Long & Sun, 2001; Bergholm et al., 2005; Plesch & Brukner, 2011), which can undermine the exponential speed-up offered by quantum algorithms. Additionally, current quantum hardware—classified as noisy intermediate-scale quantum (NISQ) devices (Preskill, 2018)—has limitations in gate fidelity, with

error rates around $\mathcal{O}(10^{-4})$ and $\mathcal{O}(10^{-3})$ for single- and two-qubit gates, respectively[1]. Consequently, it is imperative to explore strategies for generating approximate amplitude encodings (AAEs) with fewer gates, especially two-qubit gates. Recent work (Zoufal et al., 2019; Ran, 2020; Iaconis & Johri, 2023; Jobst et al., 2023; Nakaji et al., 2022; Mitsuda et al., 2024; Shirakawa et al., 2021; Rudolph et al., 2023) on AAE, based on tensor networks (Schollwöck, 2011), has introduced encoding circuits consisting of two-qubit unitaries. Although these methods highlight the relationship between small quantum entanglement and easy-to-encode states, they do not optimize circuit architectures based on the entanglement property. Furthermore, these approaches inefficiently utilize elementary two-qubit gates such as CNOT and CZ, as the decomposition of general two-qubit unitaries requires three CNOT gates (Vatan & Williams, 2004).

In this work, we introduce a novel state preparation algorithm for approximate amplitude encoding based on entropy reduction (ER-AAE). The ER-AAE algorithm for a target normalized vector $\boldsymbol{v}$ in $\mathbb{C}^{2^N}$ consists of two stages. The first stage constructs a gate sequence $G_1, G_2, \cdots, G_C$ inductively, reducing the quantum entanglement in the state that corresponds to the vector $G_C \cdots G_1 \boldsymbol{v}$ until it becomes approximately unentangled. In the second stage, a shallow gate structure $W$, involving $\mathcal{O}(N)$ one-qubit gates, is applied. Parameters in $W$ and $\{G_c\}_{c=1}^C$ are optimized to minimize the infidelity $1 - |\boldsymbol{0}^T \cdot W G_C \cdots G_1 \boldsymbol{v}|^2$. The final gate sequence for AAE is constructed as $W^\dagger, G_C^\dagger, \cdots, G_1^\dagger$. As AAE is generally a hard problem, we aim to identify conditions that guarantee the accuracy of ER-AAE. To this end, we derive an upper bound for the infidelity between the target state and the ER-AAE-generated state, which scales as $\mathcal{O}(L)$, where $L$ is the linear entropy of intermediate states with vector formulations $G_C \cdots G_1 \boldsymbol{v}$. We validate ER-AAE through state preparation tasks on synthetic datasets of Gaussian random vectors and real-world image datasets, including MNIST and CIFAR-10. Our results demonstrate that ER-AAE outperforms the best existing AAE methods in all tasks, achieving lower infidelity and higher peak signal-to-noise ratio with fewer or equal numbers of quantum gates. Additionally, we observe that real-world images exhibit a more rapid decay of linear entropy compared to random vectors, which may inspire future AAE research.

## 2 PRELIMINARY

### 2.1 NOTATIONS AND QUANTUM COMPUTING BASICS

We denote by $[N]$ the set $\{1, \cdots, N\}$. The form $\| \cdot \|_2$ represents the $\ell_2$ norm for the vector and the spectral norm for the matrix, respectively. We denote by $a_j$ the $j$-th component of the vector $\boldsymbol{a}$. The tensor product operation is denoted as "$\otimes$". The conjugate transpose of a matrix $A$ is denoted as $A^\dagger$. The trace of a matrix $A$ is denoted as $\text{Tr}[A]$. The notation $\lfloor x \rfloor$ denotes the largest integer that is smaller than or equal to $x$. We employ $\mathcal{O}$ to describe complexity notions. We employ $\tilde{\mathcal{O}}$ to describe complexity notions neglecting minor terms.

Now we introduce quantum computing knowledge and notations. The pure state of a qubit could be written as $|\phi\rangle = a|0\rangle + b|1\rangle$, where $a, b \in \mathbb{C}$ satisfy $|a|^2 + |b|^2 = 1$, and $|0\rangle = (1, 0)^T, |1\rangle = (0, 1)^T$. The $N$-qubit space is formed by the tensor product of $N$ single-qubit spaces. For pure states, the corresponding density matrix is defined as $\rho = |\phi\rangle\langle\phi|$, in which $\langle\phi| = (|\phi\rangle)^\dagger$. We use the density matrix to represent general mixed quantum states, i.e., $\rho = \sum_k c_k |\phi_k\rangle\langle\phi_k|$, where $c_k \in \mathbb{R}$ and $\sum_k c_k = 1$. A single-qubit operation to the state behaves like the matrix-vector multiplication and can be referred to as the gate —☐— in the quantum circuit language. Specifically, single-qubit operations are often used as $R_X(\theta) = e^{-i\theta X/2}$, $R_Y(\theta) = e^{-i\theta Y/2}$, and $R_Z(\theta) = e^{-i\theta Z/2}$, where

$$X = \begin{bmatrix} 0 & 1 \\ 1 & 0 \end{bmatrix}, Y = \begin{bmatrix} 0 & -i \\ i & 0 \end{bmatrix}, Z = \begin{bmatrix} 1 & 0 \\ 0 & -1 \end{bmatrix}.$$

Moreover, two-qubit operations, such as CNOT and CZ gates, are employed for generating quantum entanglement:

$$\text{CNOT} = \begin{pmatrix} 1 & 0 & 0 & 0 \\ 0 & 1 & 0 & 0 \\ 0 & 0 & 0 & 1 \\ 0 & 0 & 1 & 0 \end{pmatrix} = \quad , \quad \text{CZ} = \begin{pmatrix} 1 & 0 & 0 & 0 \\ 0 & 1 & 0 & 0 \\ 0 & 0 & 1 & 0 \\ 0 & 0 & 0 & -1 \end{pmatrix} = \quad .$$

---

[1]Data from IBM quantum devices available at https://quantum.ibm.com/services/resources.

We could obtain information from the quantum system by performing measurements, for example, measuring the state $|\phi\rangle = a|0\rangle + b|1\rangle$ generates 0 and 1 with probability $p(0) = |a|^2$ and $p(1) = |b|^2$, respectively. Such a measurement operation could be mathematically referred to as calculating the average of the observable $O = \sigma_3$ under the state $|\phi\rangle$:

$$\langle\phi|O|\phi\rangle \equiv \text{Tr}[\sigma_3|\phi\rangle\langle\phi|] = |a|^2 - |b|^2 = p(0) - p(1).$$

## 2.2 RELATED WORK

Quantum circuits for amplitude encoding have been extensively explored in prior research. Early studies (Grover, 2000; Kaye & Mosca, 2001; Long & Sun, 2001; Bergholm et al., 2005; Plesch & Brukner, 2011) demonstrated that amplitude encoding of an $N$-qubit state can be achieved using $\tilde{\mathcal{O}}(2^N)$ single- and two-qubit gates without prior knowledge. Ref. (Sun et al., 2023) introduced the use of auxiliary qubits to reduce circuit depth, though the gate count still scales exponentially. These results align with the lower bounds of state complexity. Since an $N$-qubit state represents a normalized vector in $\mathbb{C}^{2^N}$ with $2^{N+1} - 1$ degrees of freedom, it is essential to employ $\mathcal{O}(2^N)$ gates for arbitrary state preparation. Subsequent work (Gleinig & Hoefler, 2021) proposed an algorithm that constructs amplitude encoding for $S$-sparse vectors using $\mathcal{O}(SN)$ CNOT gates. However, many input states derived from classical datasets, such as real-world images (Lecun et al., 1998; Krizhevsky et al., 2009), are not sparse.

Given the inefficiency of exact amplitude encoding, researchers have focused on approximating the amplitude encoding of the target state. The central concept is to achieve this with a modest number of gates, i.e., approximate amplitude encoding trades precision for reduced quantum resources. For instance, quantum generative adversarial networks (Zoufal et al., 2019) have been used to generate quantum states from implicit data distributions. Due to the hradness of amplitude encoding for general states, it is crucial to investigate properties of easily encodable states. The simplest example is the tensor product of single-qubit states, which can be prepared from the zero state $|0\rangle$ by applying a series of single-qubit unitaries to each qubit. These tensor product states exhibit zero quantum entanglement, which is measured by the linear entropy (Manfredi & Feix, 2000) of single-qubit subsystem from the entire state. Other easily encodable states, such as those from shallow circuits, also have relatively low entanglement compared to states from deep circuits (Dankert et al., 2009). Thus, a natural approach is to assess the entanglement of the target state and employ low-entanglement approximations as an initial guess for the encoding state. This idea has catalyzed the development of a series of AAE methods based on tensor networks (Schollwöck, 2011).

Early AAE methods focused on target states with limited entanglement. Specifically, an $N$-qubit matrix product state (MPS) (Perez-Garcia et al., 2007) with bond dimension 2 can be exactly prepared using $N - 1$ two-qubit unitaries(Schön et al., 2005). This process can be sequentially applied to general quantum states (Ran, 2020), where MPS approximations are obtained via truncated singular value decomposition (SVD) (Schollwöck, 2011). More recent MPS-based AAE methods have been developed for real-world datasets (Iaconis & Johri, 2023; Jobst et al., 2023; Nakaji et al., 2022; Mitsuda et al., 2024) and specific distributions (Holmes & Matsuura, 2020; Iaconis et al., 2024). Additionally, a recent study (Shirakawa et al., 2021) proposed the AQCE algorithm for generating AAE, where locally optimal two-qubit unitaries are iteratively updated during optimization. Subsequent research (Rudolph et al., 2023) enhanced the AQCE algorithm by using an MPS as the initial guess, demonstrating superior performance compared to using the identity operator as the initial guess. Besides, several recent works (Nakaji et al., 2022; Mitsuda et al., 2024) have proposed to use fixed quantum circuit architectures such as hardware efficient ansatzes with tunable parameters as the candidate of encoding circuits, and the AAE is obtained by training parameters for minimizing the infidelity.

## 3 APPROXIMATE AMPLITUDE ENCODING BASED ON ENTROPY REDUCTION

In this section, we introduce the proposed ER-AAE in detail. The whole algorithm is an entirely classical framework that contains two components, i.e., a procedure to generate a circuit for reducing the state entanglement and a procedure to optimize the parameters in the circuit. Our framework is inspired by the fact that a quantum state with small quantum entanglement, e.g. the tensor product of 1-qubit states, can be efficiently prepared with shallow circuits. Therefore, an AAE circuit could

Figure 1: Two-qubit unitary $G$ in Algorithm 1, which contains 4 parameters.

---

**Algorithm 1** Circuit generation based on entropy reduction (ERCG)

---

**Require:** Normalized target vector $\boldsymbol{v}_{\text{target}} \in \mathbb{C}^{2^N}$, two-qubit gate set $\mathcal{S}$, two-qubit gate number threshold $C$, gate slide $C_{ER}$.
**Ensure:** Gate list $\mathcal{G} = (G_1, \cdots, G_C)$ such that the linear entropy in Eq. (2) of the vector $G_C G_{C-1} \cdots G_1 \boldsymbol{v}_{\text{target}}$ approximates to 0.
 1: Initialize $\mathcal{G} = \varnothing$, $\boldsymbol{v}_0 = \boldsymbol{v}_{\text{target}}$.
 2: **for** $i = 1$ to $C$ **do**
 3:     **for** $G \in \mathcal{S}$ **do**
 4:         Minimize the linear entropy in Eq. (2) of the vector $G\boldsymbol{v}_{i-1}$ by tuning parameters in $G$.
 5:         Record the optimized $LE_G$ and $G$.
 6:     **end for**
 7:     Find the optimal $G^* = \arg\min_G LE_G$.
 8:     **if** the qubit pair of $G^*$ has been repeated in previous $(i-1, i-2, i-3)$ iterations **then**
 9:         Choose the suboptimal gate with different qubit pairs as $G^*$.
10:     **end if**
11:     Let the new gate be $G_i = G^*$.
12:     Update the vector $\boldsymbol{v}_i = G_i \boldsymbol{v}_{i-1}$ and the gate list $\mathcal{G} = \mathcal{G} + G_i$.
13:     **if** $i\%C_{ER} = 0$ **then**
14:         Minimize the linear entropy in Eq. (2) of the vector $G_i G_{i-1} \cdots G_1 \boldsymbol{v}_{\text{target}}$ by tuning parameters in $G_1, \cdots, G_i$.
15:         Update the gate list $\mathcal{G} = (G_1, \cdots, G_i)$ with optimal parameters.
16:         Update the current vector $\boldsymbol{v}_i = G_i \cdots G_1 \boldsymbol{v}_0$.
17:     **end if**
18: **end for**
19: **return** Result

---

be obtained if we can reduce the entanglement in the quantum state gradually by adding new gate structures. The efficiency of this AAE approach would depends on the decay rate of the entanglement. Specifically, we use the sum of linear entropy of the 1-qubit subsystems from the quantum state as a metric of quantum entanglement.

The circuit generation procedure for reducing the entanglement is shown in Algorithm 1. In general, the circuit architecture is obtained via the greedy search. Firstly, we initialize the current state $|\boldsymbol{v}_0\rangle = |\boldsymbol{v}_{\text{target}}\rangle$ and the gate list $\mathcal{G} = \varnothing$. We assume that a feasible two-qubit candidate set $\mathcal{S}$ is given. Each candidate $G$ consists of four single-qubit rotations followed by one CZ gate that act on arbitrary qubit pairs,

$$\mathcal{S} = \left\{ \mathrm{CZ}(n_1, n_2) R_Y R_Z \otimes R'_Y R'_Z \big| n_1, n_2 \in [N], \ n_1 \neq n_2 \right\}. \tag{1}$$

The gate formulation in $\mathcal{S}$ is generated to employ CZ gates efficiently considering the equivalence of gate structures measured by the linear entropy as shown in Fig. 1. Specifically, we begin from the structure $V \otimes V' \mathrm{CZ} U \otimes U'$ that contains only one CZ gate. Single-qubit unitaries $V$ and $V'$ could be removed since they do not affect the value of linear entropy. Next, arbitrary single-qubit unitaries $U$ and $U'$ can be decomposed into formulations $R_Z R_Y R_Z$ and $R'_Z R'_Y R'_Z$. Since $R_Z$ and $R'_Z$ commute with the CZ gate, they can be further removed by considering the equivalence of gate structures. Therefore, we obtain the structure $G$ that contains only four single-qubit rotations.

Next, we perform the "add gate" procedure for $C$ times, where one two-qubit unitary is attached to the circuit per iteration. Different from the previous AAE methods that using two-qubit unitaries with two or more CNOT/CZ gates, the employed two-qubit unitary here contains only one CZ gate. During the $i$-th iteration, we perform a greedy search for the optimal two-qubit gate. For each feasible two-qubit gate $G$ in the gate set $\mathcal{S}$, we consider finding parameters with the lowest entanglement by

calculating the minimum of the linear entropy of the vector $\boldsymbol{v} = G\boldsymbol{v}_{i-1}$ in Eq. (2) classically:

$$LE\left(\boldsymbol{v}\right) = \sum_{n=1}^{N} \left\{ 1 - \text{Tr}_{\{n\}} \left[ \left( \text{Tr}_{[N]-\{n\}} |\boldsymbol{v}\rangle\langle\boldsymbol{v}| \right)^2 \right] \right\}. \tag{2}$$

The optimization of the linear entropy focuses on the training of 4 parameters in $G$, which is computationally efficient since each $G$ could only affect the linear entropy of two qubits. In practice, we perform this optimization using the BFGS optimizer. After the optimization with all gates in $\mathcal{S}$, we select the one with the optimal minimized linear entropy value. We remark that an arbitrary two-qubit unitary can be decomposed using 3 CNOT/CZ gates. So it could lead to redundant gates if two-qubit gates acting on the same qubits have been selected consequently for more than 3 times, even when they are locally optimal choices. For this case, we seek to the suboptimal two-qubit gates with different qubit pairs. Next, we add the selected gate as $G_i$ into the gate list $\mathcal{G}$ and update the current vector $\boldsymbol{v}_i = G_i \boldsymbol{v}_{i-1}$.

The locally optimal greedy search of $G$ may lead to suboptimal global performance. To mitigate this issue, we periodically train the whole circuit for minimizing the linear entropy in Eq. (2) after every $C_{ER}$ two-qubit gates marked as the gate slide. Finally, we generate a gate list $\mathcal{G} = (G_1, \cdots, G_C)$, such that the vector

$$\boldsymbol{v}_C = G_C \cdots G_1 \boldsymbol{v}_{\text{target}} \tag{3}$$

has a small linear entropy. For the ideal case, where the linear entropy is zero, the vector $\boldsymbol{v}_C$ is the tensor product of 2-dimensional vectors. Thereby we can prepare $|\boldsymbol{v}_C\rangle$ from the zero state by using one layer of single-qubit unitaries $W_1 \otimes \cdots \otimes W_N$, which means the state $|\boldsymbol{v}_{\text{target}}\rangle$ is constructed as

$$|\boldsymbol{v}_{\text{target}}\rangle = \left( G_1^{\dagger} \cdots G_C^{\dagger} \right) |\boldsymbol{v}_C\rangle = \left( G_1^{\dagger} \cdots G_C^{\dagger} \right) (W_1 \otimes \cdots \otimes W_N) |0\rangle := V(\boldsymbol{\theta})|0\rangle. \tag{4}$$

We denote by $V(\boldsymbol{\theta})$ in Eq. (4) the parameterized circuit that encodes the target state.

Generally, the optimization of the linear entropy could not reach zero. For this case, the circuit $V(\boldsymbol{\theta})$ provides an initial guess of the approximate amplitude encoding of the target state. Subsequently, we perform the second procedure of ER-AAE to further improve the precision. Namely, we training the parameter $\boldsymbol{\theta}$ in the encoding circuit to optimize the infidelity loss function

$$L_{\text{infid}}(\boldsymbol{\theta}; V, \boldsymbol{v}_{\text{target}}) := 1 - |\langle \boldsymbol{v}_{\text{target}} | V(\boldsymbol{\theta}) | 0 \rangle|^2. \tag{5}$$

We remark that parameters in $\prod_{i=1}^{C} G_C^{\dagger}$ have been tuned to reduce the linear entropy, but parameters in the single-qubit unitary layer have not been properly designed yet. Compared to naive uniform initializations, a reasonable strategy is to initialize the parameter in $\otimes_{n=1}^{N} W_n$ by using the information from $\boldsymbol{v}_C$. Specifically, we design the tensor network initialization (TN initialization) in Proposition 1, where the density matrix $\rho$ is the reduced density matrix of state $|\boldsymbol{v}_C\rangle$ on each qubit in $[N]$. Specifically, each single-qubit unitary $W_n$ can be constructed by using two rotations $R_Z R_Y$.

**Proposition 1.** *Denote $|\phi\rangle = R_Z(\alpha) R_Y(\theta) R_Z(\beta)|0\rangle$. Then the state $|\phi\rangle$ has the largest projection to the state $\rho$, i.e,*

$$\max_{\alpha, \beta, \theta} \text{Tr}\left[ |\phi\rangle\langle\phi|\rho \right] = \frac{1}{2} + \sqrt{\left( \frac{\rho_{00} - \rho_{11}}{2} \right)^2 + |\rho_{10}|^2} = \frac{1 + \sqrt{1 - 2LE(\rho)}}{2} \tag{6}$$

*with*

$$\beta^* = 0, \theta^* = \frac{\pi}{2} - \arcsin \frac{\rho_{00} - \rho_{11}}{\sqrt{4|\rho_{10}|^2 + (\rho_{00} - \rho_{11})^2}}, \alpha^* = \arg(\rho_{10}). \tag{7}$$

The TN initialization guarantees non-vanishing projection of the initial encoding state on the target state as shown in Proposition 2. For example, suppose the linear entropy is optimized to be $< 0.5$, the fidelity between the initial and the target state could be larger than $0.5$. Therefore, the optimization of $\boldsymbol{\theta}$ is free from the barren plateau (McClean et al., 2018) since the initial fidelity is away from zero. Proof of Propsitions 1 and 2 can be checked in the appendix. In practice, we minimize the loss in Eq. (5) by employing the Adam optimizer. After the optimization of Eq. (5) with $L_{\text{infid}}(\boldsymbol{\theta}^*) \to 0$, we obtain the approximate amplitude encoding

$$|\boldsymbol{v}_{\text{target}}\rangle \approx e^{-i\alpha_g} V(\boldsymbol{\theta}^*)|0\rangle, \tag{8}$$

where $g$ is the global phase term that would not affect the measurement result from the encoded state.

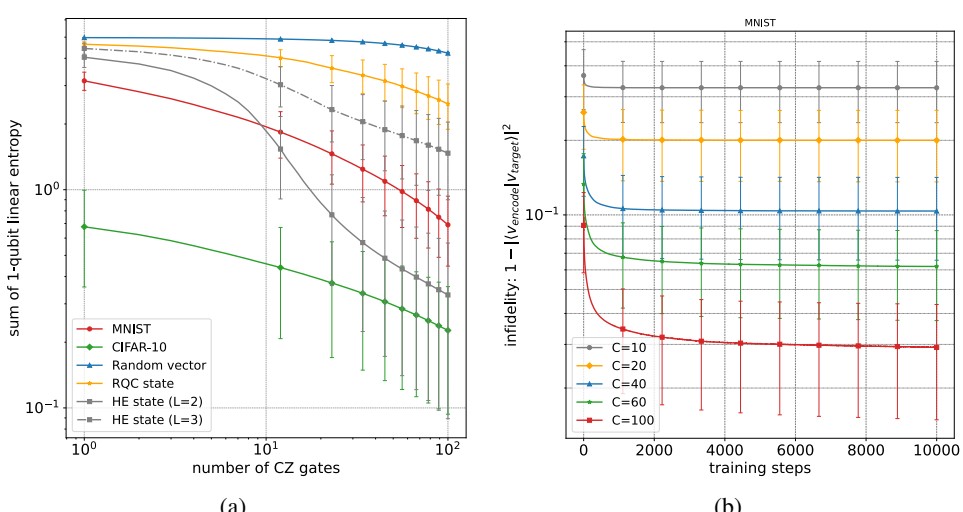

(a)                                      (b)

Figure 2: Optimization result of the ER-AAE algorithm with $T_{ER} = 0$. Fig. 2(a) shows the linear entropy during the Algorithm 1 for different datasets for the CZ gate number $C \in [100]$. Fig. 2(b) show the infidelity during the parameter optimization part of the ER-AAE algorithm with $T_{ER} = 0$ for $C \in \{10, 20, 40, 60, 100\}$. Each point is the average over $M$ independent samples, where $M = 10$ for RQC states and $M = 50$ for other cases.

Table 1: Feasible numbers of CNOT/CZ gates used in different encoding methods. The term $N$ is the number of qubits, and the term $k \in \mathbb{N}$ can be any positive integer.

|  | number of CNOT/CZ gates |
| --- | :---: |
| ER-AAE | $n_2 = k$ |
| AQCE ($\mathbb{C}$) | $n_2 = 3k$ |
| AQCE ($\mathbb{R}$) | $n_2 = 2k$ |
| MPS, AQCE-MPS ($\mathbb{C}$) | $n_2 = 3(N-1)k$ |
| MPS, AQCE-MPS ($\mathbb{R}$) | $n_2 = 2(N-1)k$ |
| ADAPT-VQE | $n_2 = 2k$ |
| HE | $n_2 = \lceil \frac{1}{2} N k \rceil$ |

**Proposition 2.** *Denote by $L$ the linear entropy loss Eq. (2) value of the state $|\boldsymbol{v}_C\rangle$ from Algorithm 1. Then the projection of the TN initial state $V(\boldsymbol{\theta})|0\rangle$ on the target state is lower bounded as*

$$|\langle \boldsymbol{v}_{\text{target}}|V(\boldsymbol{\theta})|0\rangle|^2 \geq 2^{\lfloor -2L \rfloor}. \tag{9}$$

## 4 EXPERIMENTS

In this section, we present numerical results about the performance of the ER-AAE approach on both synthetic and real-world datasets. We use the two-qubit gate number $C = 100$ and the gate slide $C_{ER} = 1$. The optimization of linear entropy after each gate slide is conducted via the Adam optimizer with learning rate 0.01 for $T_{ER}$ steps. We consider two cases $T_{ER} \in \{0, 100\}$ denoted by ERAAE-0 and ERAAE-100, respectively. The infidelity minimization phase is trained with 10000 iterations. Besides, we compare the ER-AAE approach with several existing AAE methods. Specifically, all parameter training are achieved via Adam optimizer with learning rate 0.01.

**Matrix product state approximation (MPS).** The MPS method (Ran, 2020) is an iterative approach. Given the $N$-qubit target state $|\boldsymbol{v}_{\text{target}}\rangle$ and encoding gates $G_1, \cdots, G_{(N-1)i}$ in the $i+1$-th iteration, the method generates the MPS approximation of the state $G_{(N-1)i}^{\dagger} \cdots G_1^{\dagger} |\boldsymbol{v}_{\text{target}}\rangle$ with bond dimen-

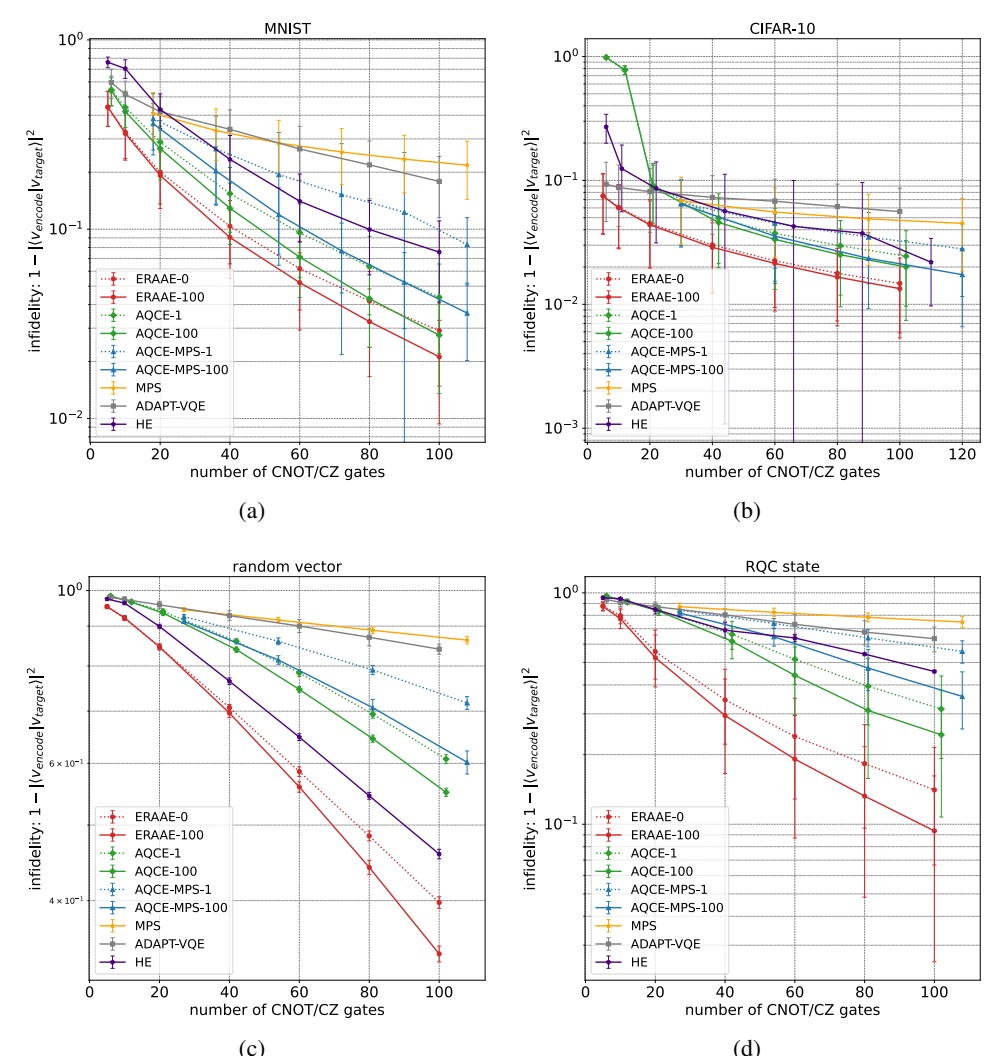

Figure 3: Result of the different AAE algorithms. Figs. 3(a)-3(d) show the infidelity corresponding to MNIST images, CIFAR-10 images, random vectors, and random quantum circuit states using different AAE algorithms. Each point is the average over $M$ independent samples, where $M = 10$ for RQC states and $M = 50$ for other cases.

sion 2. This MPS could induce a list of $N - 1$ two-qubit unitaries $G_{(N-1)i+1}, \cdots, G_{(N-1)(i+1)}$ that exactly encodes the MPS state, which are are then attached to encoding gates.

**Automatic quantum circuit encoding (AQCE).** This method was firstly proposed in Ref. (Shirakawa et al., 2021) and then being subsequently explored for constructing quantum machine learning datasets (Placidi et al., 2023). The AQCE algorithm performs locally optimal two-qubit unitary updations based on SVD. Suppose a current encoding unitary list $\{G_1, \cdots, G_M\}$ is given as the AAE of target state $|\boldsymbol{v}_{\text{target}}\rangle$ from the zero state. Then the $m$-th locally optimal two-qubit unitary is updated as

$$F_m = \text{Tr}_{[N]/\mathcal{Q}_m}\left[G_{m+1}^\dagger \cdots G_M^\dagger |\boldsymbol{v}_{\text{target}}\rangle\langle 0|G_1^\dagger \cdots G_{m-1}^\dagger\right],$$
$$X, D, Y = SVD(F_m),$$
$$G_m{}' = XY.$$

The two-qubit unitary updation is performed sequentially in a forward-backward manner for several times before adding new gates into the circuit with the same procedure. In the experiment, we consider

the number of forward-backward procedures as 1 and 100 and denote corresponding methods by AQCE-1 and AQCE-100, respectively.

**AQCE initialized with MPS approximations (AQCE-MPS)**. This procedure (Rudolph et al., 2023) considers to initialize unitaries that are added into circuits via the MPS approach. Suppose a list of unitaries are given as $|v_{\text{encode}}\rangle = G_M \cdots G_1|0\rangle$ as the AAE of $|v_{\text{target}}\rangle$ in the AQCE algorithm. Then the method calculate the new $N-1$ two-qubit unitaries via the MPS approach. Similar to the AQCE case, we consider different numbers of forward-backward procedures denoted by AQCE-MPS-1 and AQCE-MPS-100, respectively.

**ADAPT-VQE**. The method (Grimsley et al., 2019) generates AAE iteratively in a greedy search approach. In each iteration with target state $|v_{\text{target}}\rangle$ and encoded state $|v_i(\boldsymbol{\theta})\rangle = G_i(\theta_i) \cdots G_1(\theta_1)|0\rangle$, one gate $G_{i+1}(\theta_{i+1})$ is selected from the given gate set such that the fidelity $|\langle v_{\text{target}} G_{i+1}|v_i(\boldsymbol{\theta})\rangle|^2$ has the largest gradient at $\theta_{i+1} = 0$. Then the whole parameter $\boldsymbol{\theta} = (\theta_1, \cdots, \theta_{i+1})$ is updated to maximize the fidelity $|\langle v_{\text{target}}|v_{i+1}(\boldsymbol{\theta})\rangle|^2$, where $|v_{i+1}(\boldsymbol{\theta})\rangle = G_{i+1}(\theta_{i+1})|v_i(\boldsymbol{\theta})\rangle$. In the experiment, we use the gate set $\{R_X, R_Y, R_Z, CR_X, CR_Y, CR_Z\}$ and train parameters for 1000 iterations.

**Hardware-efficient (HE) circuits**. We use this method as the baseline. The HE circuit we employed consists of several HE layers initialized with the parameter $\boldsymbol{\theta} = \mathbf{0}$. Each HE layer contains a $R_Y$ rotation layer, a $R_Z$ rotation layer, and a CNOT layer that perform CNOT gates on adjacent qubit pairs. Specifically, in the $i$-th layer, CNOT gates are applied on qubit pairs $(2n, (2n+1)\%N)$, $0 \leq 2n \leq N-1$ for $i\%2 = 0$ and $(2n+1, (2n+2)\%N)$, $0 \leq 2n \leq N-2$ for $i\%2 = 1$. In the experiment, parameters are trained with 10000 iterations.

As shown above, the number of two-qubit gates have different constraints for distinct AAE methods. We summarize feasible numbers of CNOT/CZ gates in different methods in Tab. 1. Specifically, the decomposition of arbitrary two-qubit unitary requires two and three CNOT/CZ gates for the real matrix case and the general complex case (Vatan & Williams, 2004). The decomposition of $CR_X$, $CR_Y$, and $CR_Z$ gates requires two CNOT gates (Vale et al., 2023).

Next, we introduce the dataset used in this work. For the classical dataset, we employ the MNIST hand written number dataset (Lecun et al., 1998) and the CIFAR-10 dataset (Krizhevsky et al., 2009). We remark that $28 \times 28$ MNIST images are first expanded into $32 \times 32$ as the target state with qubit number $N = 10$. CIFAR-10 images are colorful with three channels of $32 \times 32$ pixels, leading to a vector $\boldsymbol{v}$ with dimension 3072. The corresponding target state is set to be the state proportional to $\boldsymbol{v}_{:2048} + i\boldsymbol{v}_{2048:}$, i.e., the classical vector is divided into two parts and filled to the dimension 2048 for being encoded into the real and the imaginary part of quantum state with $N = 11$ qubits. Thus, we could test the performance of AAE algorithms on the compact variant of amplitude encoding (Mitsuda et al., 2024). For the quantum case, we generate three datasets that contain random complex vectors, random quantum circuit (RQC) states and hardware-efficient states, respectively. Random vectors are generated from independent Gaussian distributions $\mathcal{N}(0, 1)$ for both real and imaginary parts followed by the L2 normalization. Random quantum circuits are generate by using 150 single-qubit rotations and 50 CZ gates with random structures and parameters. To distinguish from the HE method, we prepare hardware-efficient states in the last dataset by using $L$-layered hardware-efficient circuits with $L$ random parameters, where each layer consists of one $R_X$ rotation layer, one $R_Y$ rotation layer, and one CZ layer that acts CZ gates on qubit pairs $(n, (n+1)\%N)$, $0 \leq n \leq N-1$.

## 4.1 ENTROPY REDUCTION

The performance of entropy reduction of Algorithm 1 is illustrated in Fig. 2(a). We employ Algorithm 1 for MNIST and CIFAR-10 images, random vectors, RQC states, and HE states with layer $L \in \{2, 3\}$. Specifically, we remark that the behavior of the linear entropy of MNIST and CIFAR-10 images is bounded by that of HE states with layers $L = 2$ and $L = 3$, and CIFAR-10 images contain less quantum entanglements than MNIST images measured by the linear entropy. For RQC states, the entropy decays with a relatively large speed, while random vectors show some degree of robustness in the reduction of entropy, which is consistent with the fact that random vectors are hard to encode. Since the linear entropy of the real-world data decays rapidly, we expect that the corresponding ER-AAE could be accurate and efficient. Specifically, we plot the infidelity during the parameter optimization part of ER-AAE for the MNIST dataset using different numbers of two-qubit gates. The result is shown in Fig. 2(b). The loss is away from zero at the initial time, which verifies Proposition 2.

Table 2: Infidelity of AAE by different methods on MNIST, CIFAR-10, random vector, and RQC state datasets. Each point is the average over $M$ independent samples, where $M = 10$ for RQC states and $M = 50$ for other cases.

|  | MNIST | CIFAR-10 | random vector | RQC state |
|---|---|---|---|---|
| ER-AAE-0 | $0.029 \pm 0.014$ | $0.015 \pm 0.009$ | $0.398 \pm 0.007$ | $0.141 \pm 0.074$ |
| ER-AAE-100 | $\mathbf{0.021 \pm 0.012}$ | $\mathbf{0.013 \pm 0.008}$ | $\mathbf{0.342 \pm 0.008}$ | $\mathbf{0.094 \pm 0.068}$ |
| AQCE-1 | $0.044 \pm 0.022$ | $0.024 \pm 0.015$ | $0.608 \pm 0.008$ | $0.315 \pm 0.123$ |
| AQCE-100 | $0.028 \pm 0.014$ | $0.020 \pm 0.013$ | $0.551 \pm 0.007$ | $0.244 \pm 0.136$ |
| AQCE-MPS-1 | $0.083 \pm 0.032$ | $0.028 \pm 0.017$ | $0.717 \pm 0.014$ | $0.559 \pm 0.062$ |
| AQCE-MPS-100 | $0.036 \pm 0.016$ | $0.017 \pm 0.011$ | $0.602 \pm 0.021$ | $0.357 \pm 0.099$ |
| MPS | $0.218 \pm 0.074$ | $0.045 \pm 0.026$ | $0.864 \pm 0.010$ | $0.748 \pm 0.041$ |
| ADAPT-VQE | $0.179 \pm 0.063$ | $0.056 \pm 0.031$ | $0.841 \pm 0.012$ | $0.634 \pm 0.078$ |
| HE | $0.076 \pm 0.035$ | $0.022 \pm 0.012$ | $0.459 \pm 0.006$ | $0.458 \pm 0.006$ |

Table 3: Peak signal-to-noise ratio (PSNR) of AAE by different methods on MNIST and CIFAR-10 datasets. Each point is the average over 50 independent samples.

|  | MNIST | CIFAR-10 |
|---|---|---|
| ER-AAE-0 | $29.8 \pm 6.8$ | $25.1 \pm 1.9$ |
| ER-AAE-100 | $\mathbf{31.8 \pm 7.5}$ | $\mathbf{25.6 \pm 1.9}$ |
| AQCE-1 | $26.3 \pm 6.6$ | $22.8 \pm 2.0$ |
| AQCE-100 | $28.7 \pm 7.9$ | $23.7 \pm 2.0$ |
| AQCE-MPS-1 | $22.7 \pm 4.1$ | $22.2 \pm 2.0$ |
| AQCE-MPS-100 | $26.9 \pm 5.8$ | $24.5 \pm 2.0$ |
| MPS | $18.2 \pm 3.3$ | $20.0 \pm 2.1$ |
| ADAPT-VQE | $19.1 \pm 3.3$ | $18.8 \pm 1.8$ |
| HE | $23.6 \pm 5.5$ | $23.2 \pm 1.9$ |

## 4.2 AAE ACCURACY

We compare the performance of AAE using ER-AAE and other methods introduced in Section 4, with results shown in Fig. 3 and Tabs. 2 and 3. Specifically, Tabs. 2 and 3 show the performance of ER-AAE-0 and ER-AAE-100 using 100 CZ gates. The number of CZ/CNOT gates in other methods is chosen to be the smallest value in $[100, +\infty)$ according to constraints in Tab. 1. Across the MNIST, CIFAR-10, RQC state, and random vector datasets, the proposed ER-AAE consistently outperforms other existing approaches, including MPS, AQCE, AQCE-MPS, ADAPT-VQE, and the HE baseline method, demonstrating the lowest infidelity and the highest PSNR. To illustrate the performance of ER-AAE, we show the image recover of encoded states in Fig. 4. The feature of images can be roughly identified when the number of two-qubit gates exceeds 40.

## 5 CONCLUSION

In this manuscript, we propose ER-AAE, a novel classical approach for approximate amplitude encoding of real-world data based on the entropy reduction. We demonstrate that the proposed algorithm outperforms existing AAE approaches with equal or fewer CNOT or CZ gates. Besides, we show that the real-world data like MNIST or CIFAR-10 images exhibit rapid decay on the linear entropy, which may be of independent interest that can inspire other AAE research on real-world data.

## REFERENCES

Scott Aaronson. Read the fine print. *Nature Physics*, 11(4):291–293, Apr 2015. ISSN 1745-2481. doi: 10.1038/nphys3272. URL https://doi.org/10.1038/nphys3272.

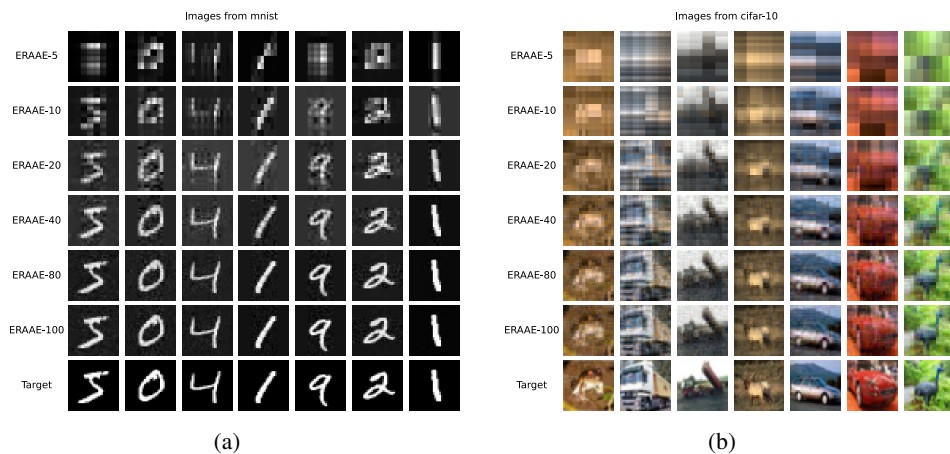

(a)                                           (b)

Figure 4: Exhibitions of the AAE of hand written numbers from the MNIST dataset and images from the CIFAR-10 dataset. Figs. 4(a) and 4(b) show the result of ER-AAE-100 using $C \in \{5, 10, 20, 40, 80, 100\}$ CZ gates.

Frank Arute, Kunal Arya, Ryan Babbush, and et al. Quantum supremacy using a programmable superconducting processor. *Nature*, 574(7779):505–510, Oct 2019. ISSN 1476-4687. doi: 10. 1038/s41586-019-1666-5. URL https://doi.org/10.1038/s41586-019-1666-5.

Ville Bergholm, Juha J. Vartiainen, Mikko Möttönen, and Martti M. Salomaa. Quantum circuits with uniformly controlled one-qubit gates. *Phys. Rev. A*, 71:052330, May 2005. doi: 10.1103/PhysRevA. 71.052330. URL https://link.aps.org/doi/10.1103/PhysRevA.71.052330.

Jacob Biamonte, Peter Wittek, Nicola Pancotti, Patrick Rebentrost, Nathan Wiebe, and Seth Lloyd. Quantum machine learning. *Nature*, 549(7671):195–202, Sep 2017. ISSN 1476-4687. doi: 10.1038/nature23474. URL https://doi.org/10.1038/nature23474.

Andrew J. Daley, Immanuel Bloch, Christian Kokail, Stuart Flannigan, Natalie Pearson, Matthias Troyer, and Peter Zoller. Practical quantum advantage in quantum simulation. *Nature*, 607 (7920):667–676, Jul 2022. ISSN 1476-4687. doi: 10.1038/s41586-022-04940-6. URL https://doi.org/10.1038/s41586-022-04940-6.

Christoph Dankert, Richard Cleve, Joseph Emerson, and Etera Livine. Exact and approximate unitary 2-designs and their application to fidelity estimation. *Phys. Rev. A*, 80:012304, Jul 2009. doi: 10.1103/PhysRevA.80.012304. URL https://link.aps.org/doi/10.1103/PhysRevA.80.012304.

Niels Gleinig and Torsten Hoefler. An efficient algorithm for sparse quantum state preparation. In *2021 58th ACM/IEEE Design Automation Conference (DAC)*, pp. 433–438, 2021. doi: 10.1109/DAC18074.2021.9586240.

Harper R. Grimsley, Sophia E. Economou, Edwin Barnes, and Nicholas J. Mayhall. An adaptive variational algorithm for exact molecular simulations on a quantum computer. *Nature Communications*, 10(1):3007, Jul 2019. ISSN 2041-1723. doi: 10.1038/s41467-019-10988-2. URL https://doi.org/10.1038/s41467-019-10988-2.

Lov K. Grover. Synthesis of quantum superpositions by quantum computation. *Phys. Rev. Lett.*, 85: 1334–1337, Aug 2000. doi: 10.1103/PhysRevLett.85.1334. URL https://link.aps.org/doi/10.1103/PhysRevLett.85.1334.

Aram W. Harrow, Avinatan Hassidim, and Seth Lloyd. Quantum algorithm for linear systems of equations. *Phys. Rev. Lett.*, 103:150502, Oct 2009. doi: 10.1103/PhysRevLett.103.150502. URL https://link.aps.org/doi/10.1103/PhysRevLett.103.150502.

Adam Holmes and A. Y. Matsuura. Efficient quantum circuits for accurate state preparation of smooth, differentiable functions. In *2020 IEEE International Conference on Quantum Computing and Engineering (QCE)*, pp. 169–179, 2020. doi: 10.1109/QCE49297.2020.00030.

Jason Iaconis and Sonika Johri. Tensor network based efficient quantum data loading of images, 2023.

Jason Iaconis, Sonika Johri, and Elton Yechao Zhu. Quantum state preparation of normal distributions using matrix product states. *npj Quantum Information*, 10(1):15, Jan 2024. ISSN 2056-6387. doi: 10.1038/s41534-024-00805-0. URL https://doi.org/10.1038/s41534-024-00805-0.

Bernhard Jobst, Kevin Shen, Carlos A. Riofrío, Elvira Shishenina, and Frank Pollmann. Efficient mps representations and quantum circuits from the fourier modes of classical image data, 2023.

Phillip Kaye and Michele Mosca. Quantum networks for generating arbitrary quantum states. In *Optical Fiber Communication Conference and International Conference on Quantum Information*, pp. PB28. Optica Publishing Group, 2001. doi: 10.1364/ICQI.2001.PB28. URL https://opg.optica.org/abstract.cfm?URI=ICQI-2001-PB28.

Iordanis Kerenidis, Jonas Landman, Alessandro Luongo, and Anupam Prakash. q-means: A quantum algorithm for unsupervised machine learning. In H. Wallach, H. Larochelle, A. Beygelzimer, and et al. (eds.), *Advances in Neural Information Processing Systems*, volume 32. Curran Associates, Inc., 2019. URL https://proceedings.neurips.cc/paper_files/paper/2019/file/16026d60ff9b54410b3435b403afd226-Paper.pdf.

Alex Krizhevsky, Geoffrey Hinton, et al. Learning multiple layers of features from tiny images. 2009.

Martin Larocca, Supanut Thanasilp, Samson Wang, Kunal Sharma, Jacob Biamonte, Patrick J. Coles, Lukasz Cincio, Jarrod R. McClean, Zoë Holmes, and M. Cerezo. A review of barren plateaus in variational quantum computing, 2024. URL https://arxiv.org/abs/2405.00781.

Y. Lecun, L. Bottou, Y. Bengio, and P. Haffner. Gradient-based learning applied to document recognition. *Proceedings of the IEEE*, 86(11):2278–2324, 1998. doi: 10.1109/5.726791.

Jin-Peng Liu, Herman Oie Kolden, Hari K. Krovi, Nuno F. Loureiro, Konstantina Trivisa, and Andrew M. Childs. Efficient quantum algorithm for dissipative nonlinear differential equations. *Proceedings of the National Academy of Sciences*, 118(35):e2026805118, 2021. doi: 10.1073/pnas.2026805118. URL https://www.pnas.org/doi/abs/10.1073/pnas.2026805118.

Gui-Lu Long and Yang Sun. Efficient scheme for initializing a quantum register with an arbitrary superposed state. *Phys. Rev. A*, 64:014303, Jun 2001. doi: 10.1103/PhysRevA.64.014303. URL https://link.aps.org/doi/10.1103/PhysRevA.64.014303.

Michael Lubasch, Jaewoo Joo, Pierre Moinier, Martin Kiffner, and Dieter Jaksch. Variational quantum algorithms for nonlinear problems. *Phys. Rev. A*, 101:010301, Jan 2020. doi: 10.1103/PhysRevA.101.010301. URL https://link.aps.org/doi/10.1103/PhysRevA.101.010301.

Lars S. Madsen, Fabian Laudenbach, Mohsen Falamarzi. Askarani, Fabien Rortais, Trevor Vincent, Jacob F. F. Bulmer, Filippo M. Miatto, Leonhard Neuhaus, Lukas G. Helt, Matthew J. Collins, Adriana E. Lita, Thomas Gerrits, Sae Woo Nam, Varun D. Vaidya, Matteo Menotti, Ish Dhand, Zachary Vernon, Nicolás Quesada, and Jonathan Lavoie. Quantum computational advantage with a programmable photonic processor. *Nature*, 606(7912):75–81, Jun 2022. ISSN 1476-4687. doi: 10.1038/s41586-022-04725-x. URL https://doi.org/10.1038/s41586-022-04725-x.

G. Manfredi and M. R. Feix. Entropy and wigner functions. *Phys. Rev. E*, 62:4665–4674, Oct 2000. doi: 10.1103/PhysRevE.62.4665. URL https://link.aps.org/doi/10.1103/PhysRevE.62.4665.

Jarrod R McClean, Sergio Boixo, Vadim N Smelyanskiy, Ryan Babbush, and Hartmut Neven. Barren plateaus in quantum neural network training landscapes. *Nat. Commun.*, 9(1):1–6, 2018. URL https://www.nature.com/articles/s41467-018-07090-4.

Naoki Mitsuda, Tatsuhiro Ichimura, Kouhei Nakaji, Yohichi Suzuki, Tomoki Tanaka, Rudy Raymond, Hiroyuki Tezuka, Tamiya Onodera, and Naoki Yamamoto. Approximate complex amplitude encoding algorithm and its application to data classification problems. *Phys. Rev. A*, 109:052423, May 2024. doi: 10.1103/PhysRevA.109.052423. URL https://link.aps.org/doi/10.1103/PhysRevA.109.052423.

Ashley Montanaro. Quantum algorithms: an overview. *npj Quantum Information*, 2(1):15023, Jan 2016. ISSN 2056-6387. doi: 10.1038/npjqi.2015.23. URL https://doi.org/10.1038/npjqi.2015.23.

Ashley Montanaro and Sam Pallister. Quantum algorithms and the finite element method. *Phys. Rev. A*, 93:032324, Mar 2016. doi: 10.1103/PhysRevA.93.032324. URL https://link.aps.org/doi/10.1103/PhysRevA.93.032324.

Kouhei Nakaji, Shumpei Uno, Yohichi Suzuki, Rudy Raymond, Tamiya Onodera, Tomoki Tanaka, Hiroyuki Tezuka, Naoki Mitsuda, and Naoki Yamamoto. Approximate amplitude encoding in shallow parameterized quantum circuits and its application to financial market indicators. *Phys. Rev. Res.*, 4:023136, May 2022. doi: 10.1103/PhysRevResearch.4.023136. URL https://link.aps.org/doi/10.1103/PhysRevResearch.4.023136.

D. Perez-Garcia, F. Verstraete, M. M. Wolf, and J. I. Cirac. Matrix product state representations. *Quantum Info. Comput.*, 7(5):401–430, jul 2007. ISSN 1533-7146.

Leonardo Placidi, Ryuichiro Hataya, Toshio Mori, Koki Aoyama, Hayata Morisaki, Kosuke Mitarai, and Keisuke Fujii. Mnisq: A large-scale quantum circuit dataset for machine learning on/for quantum computers in the nisq era, 2023.

Martin Plesch and Časlav Brukner. Quantum-state preparation with universal gate decompositions. *Phys. Rev. A*, 83:032302, Mar 2011. doi: 10.1103/PhysRevA.83.032302. URL https://link.aps.org/doi/10.1103/PhysRevA.83.032302.

John Preskill. Quantum Computing in the NISQ era and beyond. *Quantum*, 2:79, August 2018. ISSN 2521-327X. doi: 10.22331/q-2018-08-06-79. URL https://doi.org/10.22331/q-2018-08-06-79.

Shi-Ju Ran. Encoding of matrix product states into quantum circuits of one- and two-qubit gates. *Phys. Rev. A*, 101:032310, Mar 2020. doi: 10.1103/PhysRevA.101.032310. URL https://link.aps.org/doi/10.1103/PhysRevA.101.032310.

Patrick Rebentrost, Masoud Mohseni, and Seth Lloyd. Quantum support vector machine for big data classification. *Phys. Rev. Lett.*, 113:130503, Sep 2014. doi: 10.1103/PhysRevLett.113.130503. URL https://link.aps.org/doi/10.1103/PhysRevLett.113.130503.

Manuel S Rudolph, Jing Chen, Jacob Miller, Atithi Acharya, and Alejandro Perdomo-Ortiz. Decomposition of matrix product states into shallow quantum circuits. *Quantum Science and Technology*, 9(1):015012, nov 2023. doi: 10.1088/2058-9565/ad04e6. URL https://dx.doi.org/10.1088/2058-9565/ad04e6.

Ulrich Schollwöck. The density-matrix renormalization group in the age of matrix product states. *Annals of Physics*, 326(1):96–192, 2011. ISSN 0003-4916. doi: https://doi.org/10.1016/j.aop.2010.09.012. URL https://www.sciencedirect.com/science/article/pii/S0003491610001752. January 2011 Special Issue.

C. Schön, E. Solano, F. Verstraete, J. I. Cirac, and M. M. Wolf. Sequential generation of entangled multiqubit states. *Phys. Rev. Lett.*, 95:110503, Sep 2005. doi: 10.1103/PhysRevLett.95.110503. URL https://link.aps.org/doi/10.1103/PhysRevLett.95.110503.

Maria Schuld and Nathan Killoran. Quantum machine learning in feature hilbert spaces. *Phys. Rev. Lett.*, 122:040504, Feb 2019. doi: 10.1103/PhysRevLett.122.040504. URL https://link.aps.org/doi/10.1103/PhysRevLett.122.040504.

Tomonori Shirakawa, Hiroshi Ueda, and Seiji Yunoki. Automatic quantum circuit encoding of a given arbitrary quantum state, 2021.

Xiaoming Sun, Guojing Tian, Shuai Yang, Pei Yuan, and Shengyu Zhang. Asymptotically optimal circuit depth for quantum state preparation and general unitary synthesis. *IEEE Transactions on Computer-Aided Design of Integrated Circuits and Systems*, 42(10):3301–3314, 2023. doi: 10.1109/TCAD.2023.3244885.

Rafaella Vale, Thiago Melo D. Azevedo, Ismael C. S. Araújo, Israel F. Araujo, and Adenilton J. da Silva. Decomposition of multi-controlled special unitary single-qubit gates, 2023. URL https://arxiv.org/abs/2302.06377.

Farrokh Vatan and Colin Williams. Optimal quantum circuits for general two-qubit gates. *Phys. Rev. A*, 69:032315, Mar 2004. doi: 10.1103/PhysRevA.69.032315. URL https://link.aps.org/doi/10.1103/PhysRevA.69.032315.

Nathan Wiebe, Daniel Braun, and Seth Lloyd. Quantum algorithm for data fitting. *Phys. Rev. Lett.*, 109:050505, Aug 2012. doi: 10.1103/PhysRevLett.109.050505. URL https://link.aps.org/doi/10.1103/PhysRevLett.109.050505.

Yulin Wu, Wan-Su Bao, Sirui Cao, and et al. Strong quantum computational advantage using a superconducting quantum processor. *Phys. Rev. Lett.*, 127:180501, Oct 2021. doi: 10.1103/PhysRevLett.127.180501. URL https://link.aps.org/doi/10.1103/PhysRevLett.127.180501.

Tao Xin, Shijie Wei, Jianlian Cui, Junxiang Xiao, Iñigo Arrazola, Lucas Lamata, Xiangyu Kong, Dawei Lu, Enrique Solano, and Guilu Long. Quantum algorithm for solving linear differential equations: Theory and experiment. *Phys. Rev. A*, 101:032307, Mar 2020. doi: 10.1103/PhysRevA.101.032307. URL https://link.aps.org/doi/10.1103/PhysRevA.101.032307.

Enrico Zardini, Enrico Blanzieri, and Davide Pastorello. A quantum k-nearest neighbors algorithm based on the euclidean distance estimation. *Quantum Machine Intelligence*, 6(1):23, Apr 2024. ISSN 2524-4914. doi: 10.1007/s42484-024-00155-2. URL https://doi.org/10.1007/s42484-024-00155-2.

Christa Zoufal, Aurélien Lucchi, and Stefan Woerner. Quantum generative adversarial networks for learning and loading random distributions. *npj Quantum Information*, 5(1):103, Nov 2019. ISSN 2056-6387. doi: 10.1038/s41534-019-0223-2. URL https://doi.org/10.1038/s41534-019-0223-2.

