

(a)

(b)

Figure 5: Initial loss and gradient of HE and ER-AAE-0 methods on the RQC state dataset with different numbers of qubits $N \in \{6, 8, 10, 12, 14\}$. Each point is the average over 10 independent samples.

# A ADDITIONAL EXPERIMENT RESULTS

## A.1 ER-AAE IS BARREN-PLATEAU-FREE

In AAE tasks, the barren plateau (BP) issue (McClean et al., 2018; Larocca et al., 2024) typically arises when the fidelity approaches zero with exponentially vanishing gradients as the qubit number grows. As demonstrated in Proposition 2, the initial fidelity of ER-AAE is sufficiently distant from zero, therefore we expect that the optimization in ER-AAE is BP-free and scalable. To numerically verify this statement, we conduct experiments on the encoding of random quantum circuit states with varying qubit numbers $N \in \{6, 8, 10, 12, 14\}$. The initial loss and gradient of ER-AAE, alongside those of the hardware-efficient (HE) method, are illustrated in Fig. 5. These results reveal a stark contrast between the two approaches as the number of qubits increases. For the HE method, the initial gradient decays exponentially, leading to BP. Conversely, for ER-AAE, the initial gradient remains large with increasing qubits, except in the $N = 6$ case, where the gradient decreases with increased numbers of CZ gates due to the convergence to the global minimum (i.e., initial fidelity approaching 1). Consequently, we conclude that ER-AAE is free from the BP issue and demonstrates scalability.

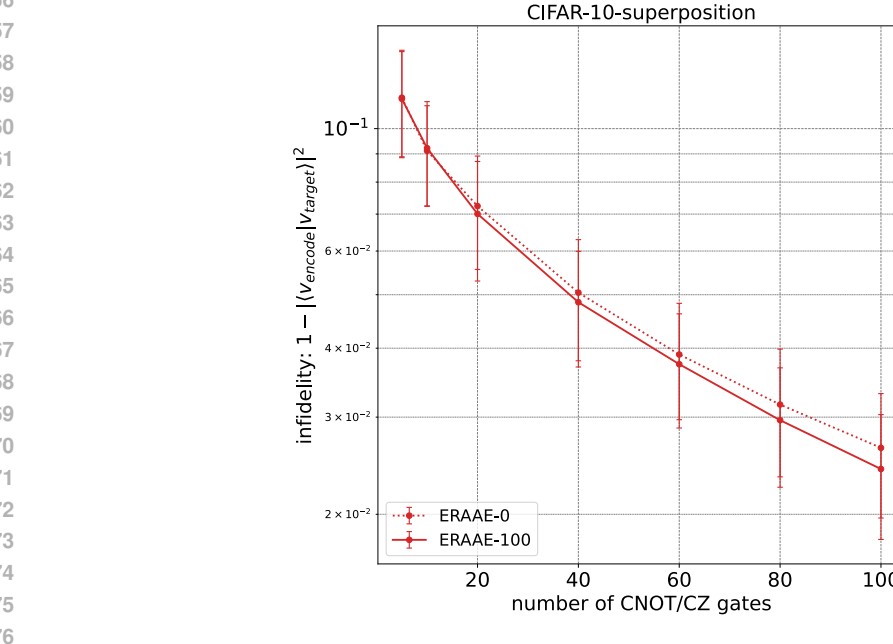

Figure 6: Infidelity of the ER-AAE encoding of the superposition of CIFAR-10 images with increased numbers of CZ gates. Each data point is the average over 10 independent samples.

## A.2 ER-AAE OF CLASSICAL DATASET IN SUPERPOSITION

Many QML algorithms require the quantum access to the classical dataset in the superposition form. Here we illustrate the performance of ER-AAE of the superposition of the CIFAR-10 dataset as an example. Specifically, each CIFAR-10 image is downsampled to $16 \times 16$ pixels. We consider the superposition of 4 images, with the result shown in Fig. 6. Similar to the single image case in the main text, the superposition of CIFAR-10 images could be encoded efficiently, achieving small infidelity below 0.03 by using 100 two-qubit gates.

## A.3 QUANTUM BINARY CLASSIFICATION WITH ER-AAE

To verify the practical usage of the proposed ER-AAE approach, we conduct the quantum binary classification on MNIST digits 1 and 3 using AAE inputs from ER-AAE and compare the results with that using exact amplitude encoding. The training minimizes the loss function defined as

$$L = \frac{1}{2|A|} \sum_{a \in A} (\text{Tr}[OV(\boldsymbol{\theta})|\boldsymbol{v}_a\rangle\langle\boldsymbol{v}_a|V(\boldsymbol{\theta})] - y_a)^2,$$

where $A$ is the training dataset, $y_a = \pm 1$ is the label. We use the 5-layered HE circuit with the structure $R_X R_Y CZ$ as the ansatz $V(\boldsymbol{\theta})$ with zero-initialized parameters. The observable is chosen to be $O = \frac{3}{N} \sum_{n=1}^{N} Z_n$. The training involves a balanced dataset that contains 8 and 6 samples in the training and the test datasets, respectively. We train parameters in $V(\boldsymbol{\theta})$ via the stochastic gradient descent with the batch size 4 and the learning rate 0.01.

Experimental results regarding to ER-AAE encoded input states with different numbers of CZ gates are shown in Fig. 7. Due to the simple structure of ER-AAE states with small numbers of CZ gates, the corresponding training achieves lower loss values compared to that of exact encoding as shown in Fig. 7(a). Other the other side, as shown in Fig. 7(b), these simple structures are sufficient to obtain similar error level compared to the exact encoding case on the test dataset.

## A.4 COMPARISON BETWEEN ER-AAE AND MPS METHODS ON THE HE STATE DATASET

The comparison between the proposed ER-AAE approach and the MPS method (Ran, 2020) is illustrated in Fig. 8. Similar to numerical results on other datasets shown in the main text, the

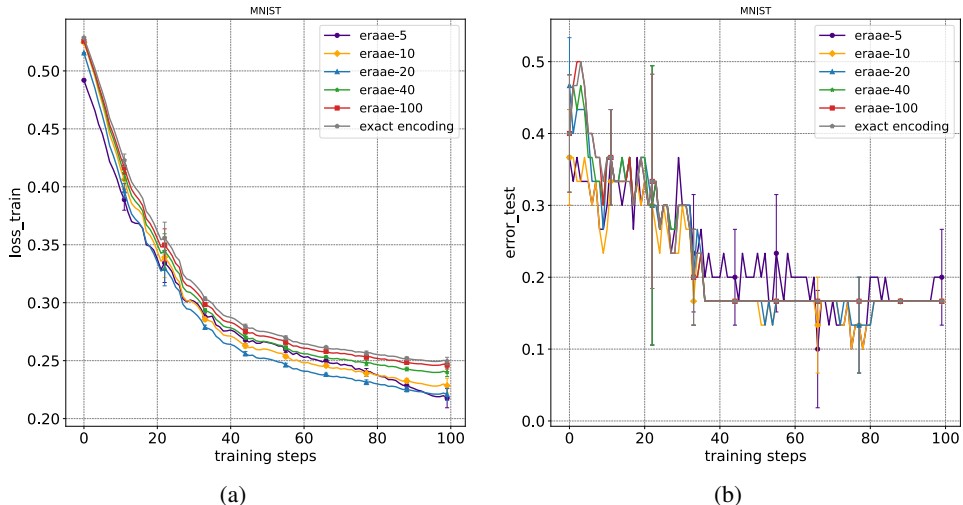

(a)      (b)

Figure 7: Quantum binary classification results of the MNIST digits 1 and 3. Figs. 7(a) and 7(b) show the training loss and the test error during the optimization. Each point is the average over $5$ independent experiments.

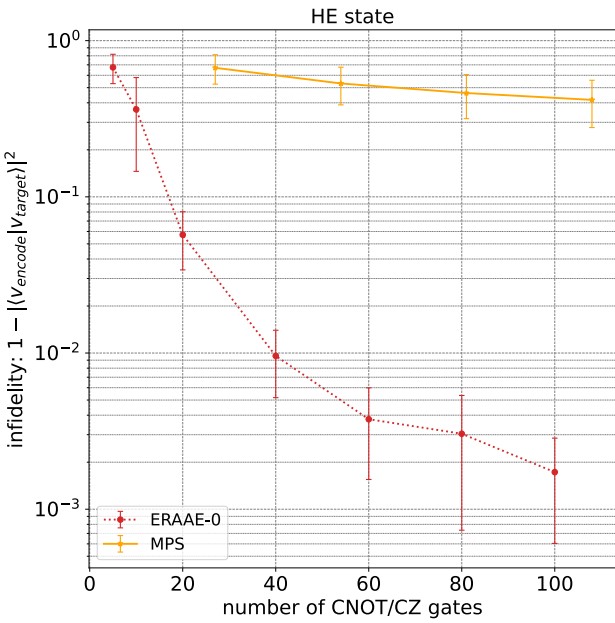

Figure 8: Infidelity of encoding with ER-AAE and MPS methods to states generated from HE circuit with $L = 2$ layers. Each data point is the average over $8$ independent samples.

ER-AAE approach significantly outperforms the MPS, achieving small infidelity values using the same or fewer numbers of CZ/CNOT gates.

## B  PROOF OF PROPOSITION 1

*Proof.* By using $R_Y(\theta) = e^{-i\theta Y/2}$ and $R_Z(\theta) = e^{-i\theta Z/2}$, we obtain

$$|\phi\rangle = e^{-\alpha Z/2} e^{-\theta Y/2} e^{-\beta Z/2} |0\rangle = \begin{bmatrix} e^{-i\frac{\alpha+\beta}{2}} \cos\frac{\theta}{2} \\ e^{i\frac{\alpha-\beta}{2}} \sin\frac{\theta}{2} \end{bmatrix}.$$

Thus, the projection

$$\text{Tr}\left[|\phi\rangle\langle\phi|\rho\right] = \rho_{00}\cos^2\frac{\theta}{2} + \left(\rho_{10}e^{-i\alpha} + \rho_{01}e^{i\alpha}\right)\sin\frac{\theta}{2}\cos\frac{\theta}{2} + \rho_{11}\sin^2\frac{\theta}{2}$$

$$= \frac{1}{2} + \frac{\rho_{00} - \rho_{11}}{2}\cos\theta + \text{Re}\left[\rho_{10}e^{-i\alpha}\right]\sin\theta, \tag{10}$$

where Eq. (10) follows from the Hermitian property $\rho = \rho^\dagger$ and $\text{Tr}\rho = 1$.

Eq. (10) is upper-bounded by

$$\text{Tr}\left[|\phi\rangle\langle\phi|\rho\right] \le \frac{1}{2} + \left|\frac{\rho_{00} - \rho_{11}}{2}\cos\theta \pm |\rho_{10}|\sin\theta\right| \tag{11}$$

$$\le \frac{1}{2} + \sqrt{\left(\frac{\rho_{00} - \rho_{11}}{2}\right)^2 + |\rho_{10}|^2}. \tag{12}$$

The equality in Eq. (11) holds when $\frac{\rho_{00} - \rho_{11}}{2}\cos\theta \pm |\rho_{10}|\sin\theta \ge 0$ and $\alpha = \arg(\rho_{10}) + 2k\pi$ ("+" case) or $\alpha = \arg(\rho_{10}) + (2k+1)\pi$ ("$-$" case). For convenience, here we consider the "+" case with $\alpha = \arg(\rho_{10})$. The equality sign in Eq. (12) holds when

$$\frac{\rho_{00} - \rho_{11}}{2} = A\sin\alpha, |\rho_{10}| = A\cos\alpha, \sin(\theta + \alpha) = 1 \tag{13}$$

with

$$A = \sqrt{\left(\frac{\rho_{00} - \rho_{11}}{2}\right)^2 + |\rho_{10}|^2}.$$

We remark that

$$\sqrt{\left(\frac{\rho_{00} - \rho_{11}}{2}\right)^2 + |\rho_{10}|^2}$$

By solving Eq. (13), we obtain one solution of $\alpha, \beta, \theta$ in Eq. (7) that yields the first part of Eq. (6). The second part of Eq. (6) can be obtained by noticing

$$LE(\rho) = 1 - \text{Tr}[\rho^2]$$
$$= 1 - \rho_{00}^2 - \rho_{11}^2 - 2|\rho_{01}|^2$$
$$= -2\rho_{00}\rho_{11} - 2|\rho_{01}|^2$$
$$= \frac{1}{2} - 2(\frac{\rho_{00} - \rho_{11}}{2})^2 - 2|\rho_{01}|^2.$$

$$\square$$

## C    PROOF OF PROPOSITION 2

*Proof.* We have

$$|\langle\boldsymbol{v}_{\text{target}}|V(\boldsymbol{\theta})|0\rangle|^2 = \left|\langle\boldsymbol{v}_{\text{target}}|G_1^\dagger\cdots G_C^\dagger\left(W_1\otimes\cdots\otimes W_N\right)|0\rangle\right|^2$$

$$= |\langle\boldsymbol{v}_C|\left(|\phi_1\rangle\otimes\cdots|\phi_N\rangle\right)|^2 \tag{14}$$

$$= \prod_{n=1}^{N}\text{Tr}\left[|\phi_n\rangle\langle\phi_n|\rho_n\right] \tag{15}$$

$$= \prod_{n=1}^{N}\frac{1 + \sqrt{1 - 2LE(\rho_n)}}{2} \tag{16}$$

$$= \frac{1}{2^N}\exp\left[\sum_{n=1}^{N}\ln(1 + \sqrt{x_n})\right]. \tag{17}$$

States $|\phi_n\rangle$ in Eq. (14) are single-qubit states generated according to the TN initialization of $|v_C\rangle$. Density matrices $\rho_n$ in Eq. (15) are the $2 \times 2$ reduced density matrix of the state $|v_C\rangle$ on the $n$-th qubit. Eq. (15) follows from tracing out qubits in $[N]$ sequentially. Eq. (16) is the direct result of the TN initialization. Eq. (17) is derived by using the notation $x_n := 1 - 2LE(\rho_n) \in [0, 1]$.

To further bond Eq. (17), we study the behavior of function $f(x) = \ln(1 + \sqrt{x}) + \ln(1 + \sqrt{a - x})$, where $0 < a < 2$ is a constant and $x$ satisfies $0 \le x \le 1, 0 \le a - x \le 1$. The derivative of $f$ is

$$f'(x) = \frac{1}{2}\left(\frac{1}{\sqrt{x}(1 + \sqrt{x})} - \frac{1}{\sqrt{a - x}(1 + \sqrt{a - x})}\right).$$

It is easy to note the $f$ monotonically increases when $x \le \frac{a}{2}$ and monotonically decreases with $x \ge \frac{a}{2}$. Thus, we have

$$f(x) \ge \begin{cases} \min(f(0), f(a)) = \ln(1 + \sqrt{a}), & \forall \, 0 < a \le 1, \\ \min(f(a - 1), f(1)) = \ln 2 + \ln(1 + \sqrt{a - 1}), & \forall \, 1 < a < 2. \end{cases} \tag{18}$$

Employing Eq. (18) in Eq. (17) with the condition $\sum_{n=1}^{N} x_n = N - 2L$, we have

$$|\langle v_{\text{target}}|V(\boldsymbol{\theta})|0\rangle|^2 \ge \frac{1}{2^N} \exp\left[\ln\left(1 + \sqrt{N - 2L - \lfloor N - 2L \rfloor}\right) + \lfloor N - 2L \rfloor \ln 2\right]$$

$$= 2^{\lfloor -2L \rfloor}\left(1 + \sqrt{N - 2L - \lfloor N - 2L \rfloor}\right)$$

$$\ge 2^{\lfloor -2L \rfloor}.$$

$\square$