# OpenReview forum: "ER-AAE: A quantum state preparation approach based on entropy reduction"
_ICLR.cc/2025/Conference — Submitted to ICLR 2025_

### Official Review · Reviewer_vE2g · 2024-10-24

**Soundness:** 3
**Presentation:** 3
**Contribution:** 2
**Rating:** 3
**Confidence:** 5

**Summary:**

This paper focuses on the amplitude encoding problem, which is one of the most significant subroutines in the context of quantum machine learning. The authors argue that general amplitude encoding (in the worst-case scenario) may require exponentially large quantum gates. In this paper, the authors introduce an Entropy Reduction AAE method to address this problem. The ER-AAE algorithm utilizes variational principles and greedy search strategies to reduce linear entropy and ultimately produce a quantum circuit. The authors numerically benchmark their method on several popular classical datasets.

**Strengths:**

From a high-level perspective, this paper is well-written, clearly introducing the research background on the amplitude encoding methods and applications, meanwhile presenting their results clearly. Meanwhile, the authors have put effort into the numerical simulation section, where they benchmark the proposed method across several datasets.

**Weaknesses:**

My main concern is about the problem setup, as outlined on Page 2 and detailed in Algorithm 1. If my understanding is correct, the authors assume that many copies of the target quantum state $|v\rangle$ are provided. Using these prepared copies of $|v\rangle$, Alg. 1  aims to find a set of quantum gates $G_1,\cdots,G_C$ (and W) to approximately prepare $|v\rangle$. The main question is: if $v$ originates from a classical dataset, where do these quantum states $|v\rangle$ come from? And if many copies of $|v\rangle$ are already available, why not use these quantum states directly to implement machine learning tasks? This setup is very different from the results in [J. Iaconis et al., npj Quantum Information, 2024], which first encode classical data into an MPS (Matrix Product State) and then transform the MPS into a quantum circuit using methods such as those proposed by [Shi-Ju Ran, Phys. Rev. A, 2020]. From this perspective, the problem statement (specifically, the requirement of Algorithm 1) seems quite strange to me. Specifically, my main concern can be summarized by:
(1) Whether they assume access to prepared quantum states or just classical data vectors
(2) If quantum states are assumed, how these are obtained from classical data
(3) How the proposed method compares to directly using prepared quantum states, in terms of efficiency and practicality for machine learning tasks

**Questions:**

1. A detailed explanation of how they evaluate linear entropy in practice
2. An analysis or estimate of the number of copies of $|v\rangle$ required to achieve a given precision in the entropy estimation
3. A discussion of how the sample complexity affects the overall efficiency of their method
4. As claimed by authors, after finding quantum gates $G_1,\cdots,G_C$, the quantum state $|v\rangle$ is mapped to a tensor-product state $|v_c\rangle$. In Eq. 5, authors mentioned that the trace distance ($L_{infid}$) can be used to train the circuit $V(\theta)$. However, it is known that the global observable may result in serious barren plateaus phenomenons [M. Cerezo et al., Nat Commun 12, 1791 (2021).] Whether this phenomenon may dramatically affect the algorithm efficiency, leading the proposed method is not scalable?
5. In proposition 2, the quantum state $|\phi\rangle$ looks like a single-qubit state. What is the relationship between $|\phi\rangle$ and Eq. 8?

---

> ### Author Response · Authors · 2024-11-26
> **Respond to Reviewer vE2g**
>
> Thanks for your constructive review! Below are our responses to the questions raised.
>
> **Q1**: A detailed explanation of how they evaluate linear entropy in practice.
>
> **R1**: We emphasize that Algorithm 1 operates entirely within a classical framework. It processes the target vector in its classical representation as input and generates a quantum gate sequence for approximate amplitude encoding. Notably, both the intermediate states and the linear entropy terms in Algorithm 1 are computed using classical computational resources. Additionally, the parameter update process following Algorithm 1 is also executed through classical methods.
>
> **Q2**: An analysis or estimate of the number of copies of |v⟩ required to achieve a given precision in the entropy estimation.
>
> **R2**: As stated above, linear entropy is computed classically rather than being estimated through quantum algorithms. Consequently, the precision of entropy computation is inherently unaffected by quantum-related uncertainties, and there is no requirement for quantum resources to address this task.
>
> **Q3**: A discussion of how the sample complexity affects the overall efficiency of their method.
>
> **R3**: As stated above, the entire ER-AAE methodology is a classical algorithm, obviating the need for quantum resources. Therefore, the problem of sample complexity does not pertain to our approach.
>
> **Q4**: As claimed by authors, after finding quantum gates G1,⋯,GC, the quantum state |v⟩ is mapped to a tensor-product state |vc⟩. In Eq. 5, authors mentioned that the trace distance (Linfid) can be used to train the circuit V(θ). However, it is known that the global observable may result in serious barren plateaus phenomenons [M. Cerezo et al., Nat Commun 12, 1791 (2021).] Whether this phenomenon may dramatically affect the algorithm efficiency, leading the proposed method is not scalable?
>
> **R4**: The proposed ER-AAE algorithm is not susceptible to the barren plateau (BP) issue. In AAE tasks, BP typically arises when the fidelity approaches zero with exponentially vanishing gradients as the qubit number grows. However, as demonstrated in Proposition 2, the initial fidelity of ER-AAE is sufficiently distant from zero. To numerically verify that ER-AAE is free from BP, we conducted experiments on the encoding of random quantum circuit states with varying qubit numbers $N = \{6, 8, 10, 12, 14\}$. The initial loss and gradient of ER-AAE, alongside those of the hardware-efficient (HE) method, are illustrated in Figure 5 in the appendix. These results reveal a stark contrast between the two approaches as the number of qubits increases. For the HE method, the initial gradient decays exponentially, leading to BP. Conversely, for ER-AAE, the initial gradient remains large with increasing qubits, except in the N=6 case, where the gradient decreases with increased numbers of CZ gates due to the convergence to the global minimum (i.e., initial fidelity approaching 1). Consequently, we conclude that ER-AAE is free from the BP issue and demonstrates scalability.
>
>
> **Q5**: In proposition 2, the quantum state |ϕ⟩ looks like a single-qubit state. What is the relationship between |ϕ⟩ and Eq. 8?
>
> **R5**: Thanks for identifying the typo! We have removed the related sentence in the revised manuscript.

---

> > ### Comment · Reviewer_vE2g · 2024-11-27
> >
> > Thanks for your response. I still have some confusion regarding your reply.
> >
> > Q1: As the authors stated, the entire Algorithm 1 is essentially a classical algorithm. Given that Algorithm 1 requires a $2^N$-sized input ($v_{target}$), all the involved operations $Gv_i$ seem to require exponentially large classical running time. As a result, the algorithm is not efficient.
> >
> > Q2: I must admit that if Algorithm 1 is a classical algorithm, then the linear entropy can be computed exactly. However, this still requires exponentially large running time. From my understanding, this algorithm is still designed with quantum devices in mind.
> >
> > Finally, could the authors comment on my main concerns outlined in the 'Weaknesses' section?

---

> > > ### Author Response · Authors · 2024-11-27
> > > **response to new questions**
> > >
> > > **Q1**: As the authors stated, the entire Algorithm 1 is essentially a classical algorithm. Given that Algorithm 1 requires a $2^N$-sized input $v_{target}$, all the involved operations $Gv_i$ seem to require exponentially large classical running time. As a result, the algorithm is not efficient.
> > >
> > > **R1**: We agree with the reviewer that the time complexity of the proposed ER-AAE algorithm scales $O(2^N)$ for target vectors with dimension $d=2^N$. Specifically, both the two-qubit gate operation $Gv_i$ and the linear entropy calculation of a vector $v$ have the time complexity $O(2^N)$. However, we mark that all approximate amplitude encoding (AAE) methods including the work [1] also require at least $O(2^N)$ running time to query all information from the classical input. Specifically, for example, the MPS method in Ref. [1] involves SVD operations to matrices with dimension $2^{N-1} \times 2$ and two-qubit gate operations to multiply the obtained encoding unitary with the current vector in each iteration. Therefore the MPS method also requires $O(2^N)$ time complexity. The time complexity of our algorithm has the optimal order to the dimension of the input vector.
> > >
> > > **Q2**: I must admit that if Algorithm 1 is a classical algorithm, then the linear entropy can be computed exactly. However, this still requires exponentially large running time. From my understanding, this algorithm is still designed with quantum devices in mind.
> > >
> > > **R2**: As stated above, the time complexity to calculate the linear entropy is $O(2^N)$, which is comparable to the standard two-qubit gate operation that commonly exists in AAE methods such as Refs. [1] and [2]. Therefore, the overall time complexity of ER-AAE has the same order to the input vector dimension compared with existing AAE approaches, and the proposed algorithm can be executed on classical computers with the similar efficiency.
> > >
> > > [1] Iaconis J, Johri S, Zhu E Y. Quantum state preparation of normal distributions using matrix product states. npj Quantum Information, 2024, 10(1): 15.
> > >
> > > [2] Shirakawa T, Ueda H, Yunoki S. Automatic quantum circuit encoding of a given arbitrary quantum state (2021). arXiv preprint arXiv:2112.14524.

---

> > > > ### Comment · Reviewer_vE2g · 2024-11-27
> > > >
> > > > Thanks for your response. However, I cannot fully agree with your comment. Actually, when the target amplitudes satisfy the area law, the MPS method would be highly efficient, and the quantum state preparation would also be efficient by using Ref. [1]. Could you please comment on whether your Alg. 1 may be efficient when the data points follow the area law?

---

> > > > > ### Author Response · Authors · 2024-11-28
> > > > > **Response to new questions**
> > > > >
> > > > > Thanks for the reply. We split the question in two here with separated responses.
> > > > >
> > > > > **Q1**: Actually, when the target amplitudes satisfy the area law, the MPS method would be highly efficient, and the quantum state preparation would also be efficient by using Ref. [1].
> > > > >
> > > > > **R1**: There is an obfuscation about the term “efficient”. Since previously raised questions by the reviewer were considering the running time complexity, we made responses where the efficiency of an algorithm is measured by its time complexity, especially regarding to the input vector dimension. Under this criterion, we concluded that the time complexity of ER-AAE has the same order to the input dimension compared to other AAE approaches. However, the term “efficient” in Q1 raised here has different meanings. We believe the reviewer means that when target amplitudes satisfy the area law, the related MPS representations are accurate with limited bond dimensions; therefore, corresponding two-qubit gate sequences have constrained depths. However, since the target amplitude is given in the classical vector formulation, the corresponding SVD for obtaining the MPS representation would still has $O(2^N)$ time complexity, thereby the efficiency measured by the running time would has exponential scalings.
> > > > >
> > > > >
> > > > > **Q2**: Could you please comment on whether your Alg. 1 may be efficient when the data points follow the area law?
> > > > >
> > > > > **R2**: Thanks for the interesting question! We expect that the ER-AAE algorithm can generate efficient encoding gate sequences for data following the area law. We have conducted Algorithm 1 (the entropy reduction algorithm) on the amplitude of 10-qubit states generated from the shallow hardware-efficient ansatz ($L=2$) with random parameters, which are known to have limited entanglement that satisfy the area law. The result is shown in Figure 2 (a), where the linear entropy decays rapidly.
> > > > >
> > > > > Additionally, we conducted new experiments to compare the proposed ER-AAE approach with the MPS method and illustrated the result in Figure 8 in the appendix. The encoding using ER-AAE shows efficient usage of two-qubit gates, where the averaged infidelity is $\leq 10^{-2}$ with $40$ CZ gates. On the other hand, the infidelity of the encoding using MPS remains large for CZ/CNOT gates $n_2 = \{27, 54, 81, 108\}$ with the value $\geq 10^{-1}$, where the number $n_2$ corresponds to $\{1,2,3,4\}$ MPS layers by Table 1 in the main text. The performance of these two methods is consistent with their behavior on other datasets illustrated in the main text (Figure 3, Tables 2 and 3). Therefore, in practice, the ER-AAE is more efficient than the MPS method when the criterion is the number of CZ/CNOT gates used in the encoding circuit.
> > > > >
> > > > > Intuitively, we speculate that the relative poor performance of the MPS method is due to its insufficient usage of CZ/CNOT gates. For example, the structure of gate sequence in MPS is fixed while that in ER-AAE is dependent on the input data, and the decomposition of each two-qubit unitary in MPS requires 3 CZ/CNOT gates. We expect that for states following the area law, the MPS method is theoretically efficient in bounds that guarantee the accuracy of MPS approximation with limited bond dimensions. However, the large constant term makes it not comparable to the ER-AAE approach in practice.

---

> > > ### Author Response · Authors · 2024-11-27
> > > **response to the weakness**
> > >
> > > **Weakness1**: My main concern is about the problem setup, as outlined on Page 2 and detailed in Algorithm 1. If my understanding is correct, the authors assume that many copies of the target quantum state are provided. …… why not use these quantum states directly to implement machine learning tasks?
> > >
> > > **R1**: As stated in previous responses, the entire ER-AAE algorithm is a classical framework. We assume the classical form instead of the quantum access to the target vector.
> > >
> > > **Weakness2**: This setup is very different from the results in [J. Iaconis et al., npj Quantum Information, 2024], which first encode classical data into an MPS (Matrix Product State) and then transform the MPS into a quantum circuit using methods such as those proposed by [Shi-Ju Ran, Phys. Rev. A, 2020]. From this perspective, the problem statement (specifically, the requirement of Algorithm 1) seems quite strange to me.
> > >
> > > **R2**: Our setup is exactly the same as that in MPS approaches mentioned by the reviewer, where the classical vector is employed as the input.
> > >
> > > **Weakness3**: Whether they assume access to prepared quantum states or just classical data vectors? If quantum states are assumed, how these are obtained from classical data?
> > >
> > > **R3**: As stated in previous responses, we do not assume quantum access to the target vector. Instead, we employ the classical data vector form as the input.
> > >
> > > **Weakness4**: How the proposed method compares to directly using prepared quantum states, in terms of efficiency and practicality for machine learning tasks?
> > >
> > > **R4**: We have added a new experiment about the performance of ER-AAE states and exactly encoded states in quantum machine learning (QML). Specifically, we considered a binary classification problem on MNIST digits with figures and the analysis presented in A.3 in appendix. In general, the time complexity of one iteration in QML training is $O((n_{input}+n_{model})n_{\theta}n_{batch})$, where $n_{input}$ is the number of quantum gates for data encoding, $n_{model}$ is the number of quantum gates in the model circuit, $n_{\theta}$ is the number of parameters in the model circuit, and $n_{batch}$ is the batch size during the training. Due to the exponential hardness of exact encoding, i. e., the exponential scaling of $n_{input}$, the corresponding gate complexity of QML could significantly exceed that using ER-AAE states. Besides, in the noiseless simulation, these two kinds of inputs exhibit similar performances on the test error. Since ER-AAE with relatively small structures are more robust against the quantum noise compared to exact encodings with exponentially large structures, we expect that ER-AAE could be employed in QML as the alternative to the exact encoding with similar or better performances.

---

> ### Comment · Reviewer_vE2g · 2024-11-28
>
> Thanks for your response and the supplemented numerical simulation. However, I do not agree with the statement 'since the target amplitude is given in the classical vector formulation, the corresponding SVD for obtaining the MPS representation would still have $2^N$ time complexity.' When the dataset satisfies the area law, **only $O(N)$ low-rank matrixes suffice to give an MPS representation**, where the running time complexity remains highly efficient. The reason is very simple: you do not necessarily require exponentially large points to compute the $L_1$-norm distance between the target vector and the MPS state. Instead, polynomial samples suffice to evaluate the total-variation distance between two vectors (guranteed by the Chebyshev inequality). As a result, when the dataset follows the area law, **the MPS-based method is computationally efficient, while the efficiency of the method proposed in this paper remains unclear (or potentially exponentially large as the authors claimed)**. Due to the proposed method not demonstrating significantly advantages compared to previous works, I have decided not to change my rating. Thanks again for the interesting discussion.

---

> > ### Author Response · Authors · 2024-11-28
> > **Response to the time complexity of ER-AAE for data following the area law**
> >
> > Thanks for the comment by the reviewer.
> >
> > We agree with the reviewer that if random algorithms based on sampling low-rank matrix approximations instead of the exact matrix multiplication are employed, one could find efficient MPS of data following the area law. **However, the proposed ER-AAE algorithm could be computationally efficient using the same method.** The procedure is very simple, we can first calculate the MPS approximation of the data. Since the entropy of the data decays during our algorithm, both the gate operation $Gv_i$ and the corresponding single-qubit lienar entropy calculation are efficient by using the MPS representation. **Therefore, under the situation such that the MPS method is computationally efficient, the ER-AAE could be efficient as well.**

---

> > ### Author Response · Authors · 2024-11-28
> > **Response to the advantage of ER-AAE compared to previous works**
> >
> > As demonstrated in Figure 3 and Tables 2&3, the proposed ER-AAE consistently outperforms existing methods significantly across all datasets including MNIST and ICFAR-10 images, the random vector dataset, and the random quantum circuit state dataset. Specifically, for image dataset, only the ER-AAE achieves the average peak signal-to-noise ratio (PSNR) $>30$ on MNIST images and the average PSNR $>25$ on CIFAR-10 images by using 100 CZ gates.

---

### Official Review · Reviewer_vgYV · 2024-10-24

**Soundness:** 3
**Presentation:** 3
**Contribution:** 2
**Rating:** 5
**Confidence:** 5

**Summary:**

The authors introduce and numerically test a method for approximately preparing target quantum states, with an eye toward using this approach for implementing an amplitude encoding scheme for quantum machine learning. To achieve this, the authors "work backwards:" given copies of the state, the algorithm applies two-qubit gates in sequence, choosing at each step the gate which minimizes the linear entropy of the state. After this procedure is repeated a number of times, the algorithm then optimizes the fidelity with the all-zero state over single-qubit rotations. The resulting approximate state preparation circuit is then the adjoint of this circuit applied to the all-zero state.

**Strengths:**

Amplitude encoding is an important building block in many quantum machine learning algorithms, and understanding when it can be approximately performed using quantum circuits of low gate complexity (relative to the Hilbert space dimension) is important for understanding when these algorithms are efficient in practice.

**Weaknesses:**

The suggested algorithm not only requires a greedy optimization over the two-qubit gates, but also performs an optimization of the fidelity over single-qubit gates; both of these optimization procedures can lead to poor optima due to the local nature of the searches. The numerical experiments are also not fully described, leaving in question the relative merits of the authors' introduced method with previous existing methods (see Questions).

The proposed technique is also not the most novel; other quantum algorithms (such as ADAPT-VQE, Nat. Commun. 10, 3007; Overlap-ADAPT-VQE, Commun. Phys. 6, 192) use a greedy method to choose gates to apply in approximate state preparation. The main distinction is that here, the authors use a different loss for each optimization step---the linear entropy rather than, e.g., the fidelity. However, the authors give little motivation as to why this choice of loss is preferable.

**Questions:**

More discussion on the performed numerics is needed to fully understand the implications of this work. For instance, for the numerics in Figure 3, how were the models "normalized" to have equivalent computational costs? I.e., were they normalized to have identical final quantum circuit depths, or was the computational cost for performing the optimization normalized to be identical, or was the number of free parameters normalized to be identical? I ask because the authors' construction uses 6 parameters for each two-qubit unitary, and the authors numerically consider unitaries up to depth 100. This is 600 parameters for a state in a Hilbert space dimension of (in the case of the MNIST data set) 1024, and thus the authors' method parameterizes a significant fraction of the full Hilbert space; I would thus expect a similarly parameterized hardware-efficient ansatz to perform similarly well to the authors' methods.

Another question is: how do these methods compare to similar greedy-state-preparation methods, such as ADAPT-VQE, where the only difference in methods is the choice of loss function? This sort of experiment would more clearly delineate what advantages the authors' methods have over preexisting methods.

---

> ### Author Response · Authors · 2024-11-26
> **Respond to Reviewer vgYV (1/2)**
>
> Thanks for your constructive review! Below are our responses to the questions raised.
>
> **Q1**: More discussion on the performed numerics is needed to fully understand the implications of this work. For instance, for the numerics in Figure 3, how were the models "normalized" to have equivalent computational costs? I.e., were they normalized to have identical final quantum circuit depths, or was the computational cost for performing the optimization normalized to be identical, or was the number of free parameters normalized to be identical?
>
> **R1**: Thank you for the thoughtful query regarding the normalization of different methods. In the experimental section, we compare different approximate amplitude encoding (AAE) methods by normalizing the number of n_2, i.e., the number of CNOT or CZ gates. It is worth noting that different AAE methods impose varying constraints on the possible values on n_2. For instance, the proposed ER-AAE approach incrementally adds one CZ gate, allowing n_2 to be any positive integer. In contrast, the automatic quantum circuit encoding (AQCE) method [1] incrementally adds two-qubit unitaries. These unitaries require two or three CNOT gates for real and complex-valued spaces, respectively, resulting in n_2=2k or n_2=3k, where k is any positive integer. Similarly, methods based on tensor networks such as MPS [2], the ADAPT-VQE approach [3], and optimizations with hardware-efficient circuits exhibit constraints on n_2. To clarify these distinctions, we have provided a comprehensive summary of n_2 values corresponding to different methods in Table 1 of the revised manuscript. In the experiment, the ER-AAE is compared with other existing methods, where the latter uses equal or slightly larger numbers of n_2. For example, in Figure 3 and Table 2, we compare ER-AAE using n_2=100 with AQCE using n_2=3k=102 when the target state is the complex random vector.
>
> **Q2**: I ask because the authors' construction uses 6 parameters for each two-qubit unitary, and the authors numerically consider unitaries up to depth 100. This is 600 parameters for a state in a Hilbert space dimension of (in the case of the MNIST data set) 1024, and thus the authors' method parameterizes a significant fraction of the full Hilbert space; I would thus expect a similarly parameterized hardware-efficient ansatz to perform similarly well to the authors' methods.
>
> **R2**: As illustrated in Figure 1 in the revised manuscript, the final RZ gate in the decomposition of single-qubit unitary U=RZRYRZ commutes with the CZ gate. Therefore it can be merged with the first RZ gate in other two-qubit gate combinations. By considering the additional RYRZ decomposition for each qubit, the entire ER-AAE encoding needs 4n_2+2N single-qubit rotations, where n_2 is the number of CZ gates and N is the number of qubits. Besides, the hardware-efficient ansatz used in both the original and the revised manuscript consists of O(4n_2)+O(1) single-qubit rotations. Based on experiment results in Figure 3 and Tables 2 and 3, the proposed ER-AAE approach significantly outperforms the hardware-efficient approach for all kinds of datasets including MNIST and CIFAR-10 images, random vectors, and random quantum circuit states.

---

> ### Author Response · Authors · 2024-11-26
> **Respond to Reviewer vgYV (2/2)**
>
> **Q3**: Another question is: how do these methods compare to similar greedy-state-preparation methods, such as ADAPT-VQE, where the only difference in methods is the choice of loss function? This sort of experiment would more clearly delineate what advantages the authors' methods have over preexisting methods.
>
> **R3**: We have included the experimental results for ADAPT-VQE with gate candidates {RX, RY, RZ, CRX, CRY, CRZ} on all qubits or qubit pairs. As presented in Figure 3 and Tables 2 and 3, ADAPT-VQE demonstrates inferior performance compared to ER-AAE across all datasets.
>
> In addition to different loss functions, ER-AAE exhibits substantial distinctions compared to other greedy approaches, such as ADAPT-VQE and AQCE, particularly in the selection of quantum gate candidates. For instance, ADAPT-VQE selects the parameterized gate with the largest gradient at the zero point during each iteration. These gate candidates equal to the identity matrix when their parameters are zero. Moreover, the two-qubit gates in these candidates must be capable of generating entanglement, necessitating a decomposition that involves at least two CZ or CNOT gates [4], as neither CZ nor CNOT can be reduced to the identity matrix using single-qubit gates alone. A similar constraint applies to the AQCE approach, where each two-qubit unitary requires at least two or three CZ/CNOT gates for real and complex cases, respectively. Thus, the superiority of ER-AAE can, in part, be attributed to its efficient utilization of CZ gates.
>
> [1] Shirakawa T, Ueda H, Yunoki S. Automatic quantum circuit encoding of a given arbitrary quantum state (2021). arXiv preprint arXiv:2112.14524.
>
> [2] Ran S J. Encoding of matrix product states into quantum circuits of one-and two-qubit gates. Physical Review A, 2020, 101(3): 032310.
>
> [3] Grimsley H R, Economou S E, Barnes E, et al. An adaptive variational algorithm for exact molecular simulations on a quantum computer. Nature communications, 2019, 10(1): 3007.
>
> [4] Vale R, Azevedo T M D, Araújo I, et al. Decomposition of multi-controlled special unitary single-qubit gates. arXiv preprint arXiv:2302.06377, 2023.

---

> > ### Comment · Reviewer_vgYV · 2024-11-27
> >
> > We thank the authors for their changes to the manuscript. The new numerical experiments greatly aid in demonstrating the utility of their approach; in particular, the authors have successfully justified why their approach should be used over other greedy methods such as ADAPT-VQE (in that the authors' approach optimizes the number of, e.g., CZ gates, as there is no constraint that at initialization the introduced gate is equal to the identity). I have thus raised my score accordingly.

---

> ### Comment · Area_Chair_Mw3V · 2024-11-27
>
> Dear Reviewer,
>
> The authors have provided their rebuttal to your comments/questions. Given that we are not far from the end of author-reviewer discussions, it will be very helpful if you can take a look at their rebuttal and provide any further comments. Even if you do not have further comments, please also confirm that you have read the rebuttal. Thanks!
>
> Best wishes,
> AC

---

### Official Review · Reviewer_Bnz9 · 2024-11-03

**Soundness:** 2
**Presentation:** 3
**Contribution:** 2
**Rating:** 5
**Confidence:** 5

**Summary:**

The authors propose a new method for finding quantum circuits to prepare a target quantum state. The proposed algorithm starts with a target state and iteratively finds the best parametrized two qubit operations one can apply to the state. The aim to to construct such a circuit that minimizes a sum of subsystem linear entropies. The authors test their algorithm using classical simulations and show that this can be used to encode elements from various machine learning datasets into a quantum circuit with good accuracy and favorable performance.

**Strengths:**

This addresses an important problem at the intersection of machine learning and quantum computing, i.e. how to prepare quantum states that effectively encode large datasets. The paper is very clearly written and the arguments of the authors are easy to follow.

**Weaknesses:**

See below for a list of weaknesses and associated questions.

**Questions:**

There are general questions/concerns I have about the paper.
1. Could the authors clarify the setting of Algorithm 1? Is this intended as a quantum algorithm or is it meant to be simulated on a classical computer?  In other words, does the algorithm require any quantum resources?

2. In QML literature, the notion of a QRAM is often used to circumvent the data loading problem[3]. Conceptually, should I think of AR-EEE as a candidate algorithm for a QRAM? Or is it independent of the QRAM idea all together?

3. In Proposition 2, the state $| \phi \rangle$ is defined but not used anywhere in the statement.

4. If a large enough C is used, is the algorithm always guaranteed to find a $v_c$ that is a product state? Will it find such a $v_c$ for a large enough C if the assume that the greedy search and the angle optimizations used in the algorithm  never get stuck in local minima?

5. Do the authors have any intuition on why the algorithm performs poorly on random vectors in Fig 2 (a)? Is this due to limited expressivity of the variational circuit used in this procedure?

6. The authors claim that states that AR-EEE encodes well are those with low entanglement. Could the authors clarify what type of entanglement structure is favorable for AR-EEE? For states with low entanglement, is AR-EEE favorable to the naive approach of preparing a tensor network (with a large enough bond dimension similar to how C was increased in the experiments here) and then preparing that directly using a quantum circuit [1] [2]? For 2D images, it is possible that the specific low-entanglement structure can be captured by a PEPS.

7. Many QML algorithms require coherent access to a dataset in superposition [3]. While the numerical demonstration here deal with single elements from CIFAR and MNIST. Do the favorable properties, like low entanglement, shown by single images imply that these superposition states can also be prepared easily?

[1] Schön, Christian, et al. "Sequential generation of entangled multiqubit states." Physical review letters 95.11 (2005): 110503.
[2] https://arxiv.org/abs/1104.1410
[3] Biamonte, Jacob, et al. "Quantum machine learning." Nature 549.7671 (2017): 195-202.

---

> ### Author Response · Authors · 2024-11-26
> **Respond to Reviewer Bnz9 (1/2)**
>
> Thanks for your constructive feedback! Below are our responses to the questions raised.
>
> **Q1**: Could the authors clarify the setting of Algorithm 1? Is this intended as a quantum algorithm or is it meant to be simulated on a classical computer? In other words, does the algorithm require any quantum resources?
>
> **R1**: Algorithm 1 is entirely classical. It accepts the target vector in classical form as input and outputs a quantum gate sequence that facilitates the approximate amplitude encoding of the vector. Specifically, the intermediate states and linear entropy terms in Algorithm 1 are computed using classical resources. Furthermore, the parameter update process following Algorithm 1 is also implemented via classical computation. Hence, the ER-AAE approach does not require any quantum resources.
>
> **Q2**: In QML literature, the notion of a QRAM is often used to circumvent the data loading problem. Conceptually, should I think of ER-AAE as a candidate algorithm for a QRAM? Or is it independent of the QRAM idea all together?
>
> **R2**: Given an input target vector, the ER-AAE approach produces a quantum gate sequence comprising single-qubit rotations and CZ gates. This sequence enables an approximate amplitude encoding for the target vector. In essence, the output of ER-AAE, expressed as a gate sequence, can be interpreted as an efficient and approximate decomposition of the unitary operation that achieves the QRAM. Thus, the ER-AAE output can serve as a substitute for QRAM in the data-loading phase.
>
> **Q3**: In Proposition 2, the state |ϕ⟩ is defined but not used anywhere in the statement.
>
> **R3**: Thanks for identifying the typo! We have removed the related sentence in the revised manuscript.
>
> **Q4**: If a large enough C is used, is the algorithm always guaranteed to find a vc that is a product state? Will it find such a vc for a large enough C if the assume that the greedy search and the angle optimizations used in the algorithm never get stuck in local minima?
>
> **R4**: This is indeed a highly intriguing and thought-provoking question. In our experiments, we did not observe the algorithm becoming stuck in local minima during the optimization process, which suggests that the approach is robust under the tested conditions. However, a rigorous theoretical guarantee that the algorithm will always converge to a product state with a sufficiently large C would require a detailed investigation of the interplay between the greedy search strategy and the optimization of angles. While our current study primarily focuses on empirical evaluations, exploring the theoretical convergence properties of the algorithm is an exciting direction for future research, including its ability to avoid local minima and its trajectory towards the global minimum.
>
> **Q5**: Do the authors have any intuition on why the algorithm performs poorly on random vectors in Fig 2 (a)? Is this due to limited expressivity of the variational circuit used in this procedure?
>
> **R5**: We expect that random vectors generally do not have efficient approximate amplitude encoding (AAE) due to their inherent complexity, regardless of the specific AAE algorithm employed. Otherwise, for the ER-AAE case, the entropy would decay rapidly as new gates are incrementally added. According to Proposition 2 in the main text, this would imply that the AAE of random vectors should have high precision on average, accompanied by small entropy terms. This result contradicts the well-established result [1] that AAE is exponentially hard for generic quantum states (or vectors). An intuitive explanation is that a normalized complex random vector in the 2^N-dimensional space has 2^{N+1}-1 independent degrees of freedom, thereby requiring an exponentially large number of quantum gates for representation.

---

> ### Author Response · Authors · 2024-11-26
> **Respond to Reviewer Bnz9 (2/2)**
>
> **Q6**: The authors claim that states that ER-AAE encodes well are those with low entanglement. Could the authors clarify what type of entanglement structure is favorable for ER-AAE? For states with low entanglement, is ER-AAE favorable to the naive approach of preparing a tensor network (with a large enough bond dimension similar to how C was increased in the experiments here) and then preparing that directly using a quantum circuit?
>
> **R6**: We did not claim that states efficiently encoded by ER-AAE are necessarily low-entanglement states. Rather, ER-AAE finds efficient encodings when the linear entropy decays rapidly, as demonstrated by Algorithm 1 and shown in Proposition 2. The identification of conditions leading to efficient encodings (or rapid entropy decay) is an intriguing and valuable avenue for future research, but it falls beyond the scope of the current study. Intuitively, we expect that the class of quantum states efficiently encoded by ER-AAE includes not only low-entanglement states but also states generated by a relatively mild number of quantum gates. Specifically, we have incorporated experiments involving states generated by random quantum circuits consisting of 150 single-qubit rotations and 50 CZ gates. Additionally, we have provided a comparison of ER-AAE with the AAE method inspired by tensor networks given in Ref. [3], an approach developed based on Ref. [2]. The corresponding results are presented in Figure 3 and Tables 2 and 3. On all datasets—including MNIST images, CIFAR-10 images, random vectors, and states from random quantum circuits—our ER-AAE approach outperforms other AAE algorithms, including the one in Ref. [3], achieving smaller infidelity using the same or fewer CNOT/CZ gates.
>
> **Q7**: For 2D images, it is possible that the specific low-entanglement structure can be captured by a PEPS.
>
> **R7**: While one-dimensional tensor networks, such as MPS, can serve as candidates for AAE, two-dimensional tensor networks like PEPS are not a natural choice, even for 2D images. This discrepancy arises from the global and relatively dense nature of amplitude encoding, which utilizes very few number of qubits. For instance, a 32×32 image can be represented by a vector in a 1024-dimensional space, and its amplitude encoding requires only log2 (1024)=10 qubits. Conversely, a PEPS state for a 32×32 image would typically involve log2 (32)=5 qubits per dimension, resulting in a total of 5×5=25 qubits, which does not align with the qubit requirements of the amplitude encoding.
>
> **Q8**: Many QML algorithms require coherent access to a dataset in superposition. While the numerical demonstration here deal with single elements from CIFAR and MNIST. Do the favorable properties, like low entanglement, shown by single images imply that these superposition states can also be prepared easily?
>
> **R8**: Thank you for the constructive suggestion! To address this point, we have conducted an additional experiment examining the encoding of superposition states of CIFAR-10 images as a representative case. The results, detailed in Figure 6 of the supplementary material, confirm that these superposition states can also be prepared efficiently, exhibiting small infidelity. This observation suggests that the advantageous properties demonstrated by individual images are preserved in their superpositions, facilitating efficient preparation in practical quantum machine learning scenarios.
>
> [1] Plesch M, Brukner Č. Quantum-state preparation with universal gate decompositions. Physical Review A, 2011, 83(3): 032302.
>
> [2] Schön, Christian, et al. "Sequential generation of entangled multiqubit states." Physical review letters 95.11 (2005): 110503.
>
> [3] Ran S J. Encoding of matrix product states into quantum circuits of one-and two-qubit gates. Physical Review A, 2020, 101(3): 032310.

---

> > ### Comment · Reviewer_Bnz9 · 2024-11-26
> >
> > Thanks you to the authors for their response.  After considering everything, I have decided not to change my score.

---

### Official Review · Reviewer_C4ho · 2024-11-11

**Soundness:** 3
**Presentation:** 3
**Contribution:** 3
**Rating:** 6
**Confidence:** 4

**Summary:**

The paper proposes approximate amplitude encoding of classical information into quantum states. The proposed method allows to leverage the precision of the encoding with the number of the gates and thus can be adapted based on requirements. The encoding is minimized based on linear entropy reduction, calculated as entanglement reduction, by adding one gate at the time from exhaustively searching the set of available gates for the one that minimizes the linear entropy the most. The results looks promising as on the MNIST and CIFAR the proposed method seems to outperform the other approaches that are used for comparison.

**Strengths:**

- The proposed approach based on the entanglement minimization is interesting and novel.

**Weaknesses:**

- The main advantage is also a weakness. While the approximate method allows to encode the states at a lesser precision with a smaller amount of gates, it can also lead to a higher cost when encoding for 100% accurate representation

**Questions:**

Is the entropy decay same for all datasets? If not how will this method be affected if for different datasets the decay will be different?

---

> ### Author Response · Authors · 2024-11-26
> **Respond to Reviewer C4ho**
>
> Thanks for your positive and constructive feedback! Below are our responses to the questions raised.
>
> **Q1**: While the approximate method allows to encode the states at a lesser precision with a smaller amount of gates, it can also lead to a higher cost when encoding for 100% accurate representation.
>
> **R1**: We appreciate the reviewer’s insightful observation. The concern raised is indeed applicable not only to our proposed ER-AAE framework but also to approximate amplitude encoding (AAE) techniques in general. Under ideal conditions, there may exist a precision threshold p0 beyond which AAE methods require more quantum gates than exact amplitude encoding, even though the latter inherently demands an exponential number of gates. However, we contend that this phenomenon should not be identified as the weakness of our work. Striving for such extreme precision is **futile** and may **degrade** the practical performance for two reasons, as discussed in the introduction's second paragraph.
>
> Firstly, similar to the exact amplitude encoding case, achieving precision beyond p0 with AAE incurs exponential gate complexity, thereby negating the exponential quantum speed-ups of most quantum machine learning (QML) algorithms leveraging classical data [1]. Consequently, it is futile to optimize the precision of AAE at such gate complexity. Secondly, given the noise in current and near-term quantum computers, both single- and two-qubit gates are prone to errors in practical scenarios. This means that increasing the gate count, particularly of two-qubit gates (which introduce more noise than single-qubit gates), could result in degraded performance. Therefore, this study prioritizes enhancing the precision of AAE with constrained gate counts, with a particular emphasis on minimizing two-qubit CZ gate usage to fit practical situations.
>
> **Q2**: Is the entropy decay same for all datasets? If not how will this method be affected if for different datasets the decay will be different?
>
> **R2**: Similar to other known AAE methods, the performance of ER-AAE is data-dependent. We do not presume that linear entropy undergoes rapid decay across all datasets and data distributions. Otherwise, the task of approximate amplitude encoding (AAE) would not be hard with exponential gate complexity for the worst case, thereby contravening the existing theoretical result [2]. For example, as shown in Figure 3, the proposed ER-AAE method exhibits better performance on structured datasets such as MNIST and CIFAR10 compared to random vectors. Generally, entropy decay proceeds slowly when the quantum state representation of data is both unstructured and highly entangled, and the corresponding encoding is hard. Nonetheless, in experiments across all kinds of datasets, ER-AAE consistently outperforms other known encoding methodologies (Figure 3, Tables 2 and 3).
>
> [1] Aaronson S. Read the fine print. Nature Physics, 2015, 11(4): 291-293.
>
> [2] Plesch M, Brukner Č. Quantum-state preparation with universal gate decompositions. Physical Review A—Atomic, Molecular, and Optical Physics, 2011, 83(3): 032302.

---

> ### Comment · Area_Chair_Mw3V · 2024-11-27
>
> Dear Reviewer,
>
> The authors have provided their rebuttal to your comments/questions. Given that we are not far from the end of author-reviewer discussions, it will be very helpful if you can take a look at their rebuttal and provide any further comments. Even if you do not have further comments, please also confirm that you have read the rebuttal. Thanks!
>
> Best wishes,
> AC

---

### Author Response · Authors · 2024-11-28

Dear Reviewers,

We would like to sincerely thank you for the time and effort you have dedicated to reviewing our paper. Your detailed comments and thoughtful insights have been invaluable in helping us refine and improve the quality of our work. We are particularly grateful for your recognition of the novelty and contribution of our research. Such acknowledgment is deeply encouraging and motivates us to further advance this manuscript.

In responses and the revised submission, we have made every effort to address the concerns and suggestions raised during the review process. We hope that our responses and the corresponding revisions comprehensively resolve the issues highlighted.

Thank you once again for your constructive feedback and support.

---

### Meta-Review · Area_Chair_Mw3V · 2024-12-07

**Metareview:**

This paper proposed a quantum state preparation approach based on entropy reduction. The method is validated with experimental results, including state preparations on random quantum circuit states, random vectors, MNIST digits, and CIFAR-10 images.

During the reviewer and AC discussion period, the reviewers reached a consensus on rejection. In particular, the reviewers agreed with two main concerns: 1) The algorithm is essentially a classical algorithm that requires exponentially large running time, which is intractable in practice. 2) The experiments are limited, in the sense that current experimental results are not at a scale to support the assertion that this algorithm will be able to efficiently load practically useful ML datasets into a quantum machine.

Therefore, the decision is to reject this paper at ICLR 2025

**Additional Comments On Reviewer Discussion:**

There were adequate discussions between the authors and the reviewers during the rebuttal period. During the reviewer and AC discussion period, the reviewers reached a consensus on rejection - see the points in the metareview.

---

### Decision · Program_Chairs · 2025-01-22

Reject